# *C9orf72* arginine-rich dipeptide repeat proteins disrupt karyopherin-mediated nuclear import

**Lindsey R Hayes[1,2]\*, Lauren Duan[3], Kelly Bowen[1,2], Petr Kalab[4†]\*, Jeffrey D Rothstein[1,2†]**

[1]Department of Neurology, Johns Hopkins University School of Medicine, Baltimore, United States; [2]Brain Science Institute, Johns Hopkins University School of Medicine, Baltimore, United States; [3]Department of Psychological and Brain Sciences, Krieger School of Arts and Sciences, Johns Hopkins University, Baltimore, United States; [4]Department of Chemical and Biomolecular Engineering, Whiting School of Engineering, Johns Hopkins University, Baltimore, United States

**Abstract** Disruption of nucleocytoplasmic transport is increasingly implicated in the pathogenesis of neurodegenerative diseases, including ALS caused by a *C9orf72* hexanucleotide repeat expansion. However, the mechanism(s) remain unclear. Karyopherins, including importin β and its cargo adaptors, have been shown to co-precipitate with the *C9orf72* arginine-containing dipeptide repeat proteins (R-DPRs), poly-glycine arginine (GR) and poly-proline arginine (PR), and are protective in genetic modifier screens. Here, we show that R-DPRs interact with importin β, disrupt its cargo loading, and inhibit nuclear import of importin β, importin α/β, and transportin cargoes in permeabilized mouse neurons and HeLa cells, in a manner that can be rescued by RNA. Although R-DPRs induce widespread protein aggregation in this in vitro system, transport disruption is not due to nucleocytoplasmic transport protein sequestration, nor blockade of the phenylalanine-glycine (FG)-rich nuclear pore complex. Our results support a model in which R-DPRs interfere with cargo loading on karyopherins.

**\*For correspondence:**
lhayes@jhmi.edu (LRH);
petr@jhu.edu (PK)

[†]These authors contributed equally to this work

**Competing interests:** The authors declare that no competing interests exist.

## Introduction

A GGGGCC hexanucleotide repeat expansion (HRE) in *C9orf72* is the most common known cause of amyotrophic lateral sclerosis (ALS) and is also a major cause of frontotemporal dementia (FTD) and the ALS/FTD overlap syndrome (*DeJesus-Hernandez et al., 2011*; *Renton et al., 2011*; *Majounie et al., 2012*). The *C9orf72* HRE is thought to cause disease by a toxic gain of function involving expanded repeat RNA and dipeptide repeat proteins (DPRs) produced by repeat-associated (non-AUG) translation, although a modest reduction in C9ORF72 protein is also seen (reviewed by *Cook and Petrucelli, 2019*). Predicted products of *C9orf72* HRE translation in both the sense (poly-GP, poly-GA, poly-GR) and antisense (poly-GP, poly-PR, poly-PA) directions have been identified in postmortem tissue (*Zu et al., 2013*; *Ash et al., 2013*; *Mackenzie et al., 2013*; *Gendron et al., 2013*), and overexpression of a subset of DPRs, including poly-GA and the arginine-containing DPRs poly-GR and poly-PR (R-DPRs), is toxic in cell culture (*May et al., 2014*; *Wen et al., 2014*) and animal models (*Zhang et al., 2016*; *Zhang et al., 2018*; *Zhang et al., 2019*).

Growing evidence suggests that disruption of nucleocytoplasmic transport (NCT), the regulated trafficking of proteins and ribonucleoprotein complexes between the nucleus and cytoplasm, is a major pathophysiologic mechanism in neurodegenerative diseases (reviewed by *Hutten and Dormann, 2019*). Bidirectional NCT across the nuclear envelope occurs through nuclear pore complexes (NPC), which are large (125 MDa) assemblies comprised of multiple copies of ~30 different

nucleoporins (Nups) (*Reichelt et al., 1990*). Although small cargoes passively equilibrate across the NPC, larger cargoes are increasingly excluded by a matrix of natively unfolded phenylalanine-glycine (FG)-rich nucleoporins lining the central channel (*Mohr et al., 2009*; *Timney et al., 2016*; *Frey et al., 2018*). Transport of restricted cargoes requires karyopherins (also known as nuclear transport receptors), including importins (importins and transportins), exportins, and bidirectional transporters that mediate the rapid transport of cargo through the FG-barrier (reviewed by *Baade and Kehlenbach, 2019*). The small GTPase Ran dictates the directionality of transport via a steep concentration gradient of RanGTP across the nuclear membrane, established by the nuclear guanine nucleotide exchange factor RCC1 and the cytoplasmic GTPase-activating protein RanGAP1. Nuclear RanGTP promotes importin-cargo unloading and exportin-cargo complex assembly, while the cytoplasmic conversion of RanGTP to RanGDP disassembles exportin-cargo complexes and enables importin-cargo binding.

We and others have found evidence of NPC and NCT disruption in postmortem tissue and animal models of *C9orf72*-ALS, Alzheimer's disease, and Huntington's disease, including mislocalization and loss of Nups and disruption of the Ran gradient (*Zhang et al., 2015*; *Grima et al., 2017*; *Eftekharzadeh et al., 2018*). The functional implications of this pathology for NCT, and consequences for neuronal survival, remain largely unknown. However, cytoplasmic aggregates of amyloid-like proteins, including artificial β-sheet proteins and poly-GA, have been reported to sequester Nups, karyopherins, and other components of the NCT machinery in cell culture models, associated with nuclear transport deficits (*Woerner et al., 2016*; *Khosravi et al., 2016*). Interactome screens have also shown that *C9orf72* R-DPRs co-precipitate NPC and NCT proteins, notably importins, including importin β, its importin α family of cargo adaptors, and transportin (*Lee et al., 2016*; *Lin et al., 2016*; *Yin et al., 2017*). *C9orf72* genetic modifier screens in *Drosophila*, yeast, and neurons have also identified a beneficial role for this class of proteins (*Zhang et al., 2015*; *Freibaum et al., 2015*; *Jovičić et al., 2015*; *Boeynaems et al., 2016*; *Kramer et al., 2018*). However, direct interactions between R-DPRs and importins, and consequences for nuclear transport, remain to be explored.

Importins represent the largest group of karyopherins (16 out of more than 20 in vertebrates) and they operate in parallel to transport a repertoire of distinct cargoes defined by their nuclear localization signals (NLS) (reviewed by *Baade and Kehlenbach, 2019*; *Oka and Yoneda, 2018*). While importin β (KPNB1) binds directly to a subset of cargoes, most cargoes are loaded on importin β via a heterodimer with importin α (KPNA2 and 6 others in humans). The C-terminal NLS-binding site of free importins α is autoinhibited by the N-terminal importin β-binding domain (IBB), and unmasked upon binding to importin β, enabling cargo recognition and formation of the trimeric import complex (cargo•importin α•importin β) (reviewed by *Lott and Cingolani, 2011*). A wide array of importin α/β cargoes have been reported (*Baade and Kehlenbach, 2019*), including TDP-43, a nuclear DNA/RNA-binding protein that mislocalizes to the cytoplasm and forms pathologic aggregates in >97% of ALS cases (*Neumann et al., 2006*; *Mackenzie et al., 2014*). Transportins-1 and -2 are a class of importins that bind cargoes directly and with considerable overlap based on the PY-NLS consensus sequence, found in cargoes such as hnRNPs and FUS (reviewed by *Soniat and Chook, 2015*). Notably, the NLSs recognized by importin α (mono- and bipartite) or directly by importin β, the PY-NLS, and the importin α IBB (a disordered region that is also a functional NLS), are all enriched in arginine and lysine residues that mediate high-affinity interactions within nuclear import complexes. We hypothesized that R-DPRs may mimic these arginine-and lysine-rich domains, binding either to the import cargoes or their carriers, and disrupting the formation of importin•cargo complexes.

Here, we took advantage of well-characterized fluorescent probes, FRET techniques and biochemical assays to investigate R-DPR-mediated effects on the importin β pathway, and show that R-DPRs bind importin β, disrupt importin β•IBB interactions, and confer dose- and length-dependent disruption of importin β-mediated nuclear import in the permeabilized cell assay. As predicted, R-DPRs also disrupt transportin-dependent nuclear import. R-DPRs rapidly induce aggregate formation within the transport assay, which recruit numerous RNA-binding and ribosomal proteins, as well as NPC and NCT proteins. However, by separating the soluble and insoluble phases of the reaction, we show that transport disruption in this model is not due to sequestration of NCT components, nor blockade of the FG-permeability barrier, but due to perturbation of importins in the vicinity of the nuclear envelope, an effect that is reversible by RNA. Notably, the behavior of GR and PR differed in many of our assays, with PR overall acting as a more potent and selective inhibitor.

## Results

### R-DPRs bind importin β and inhibit nuclear import

Although importin β has been shown to co-precipitate with poly-GR and poly-PR, a direct interaction between R-DPRs and importin β has not been demonstrated, and the consequences for functional nuclear import are unknown. To test for an interaction between *C9orf72* DPRs and importin β, we used a variant of the FRET sensor Rango ('Ran-regulated importin β cargo') (*Kaláb et al., 2006*), which consists of the IBB domain of importin α1 (KPNA2), flanked by CyPet (donor) and YPet (acceptor). When bound to importin β (KPNB1), Rango FRET is low, but in the presence of RanGTP, importin β is displaced from the sensor and FRET increases (*Figure 1A–C*). Since conserved arginine and lysine residues of the IBB domain are required for binding to importin β (*Görlich et al., 1996*; *Weis et al., 1996*; *Cingolani et al., 1999*), we hypothesized that the arginine-rich DPRs could bind to the corresponding sites on importin β and compete with the IBB. Synthetic GP10, GA10, and PA10 peptides did not affect Rango FRET even at high concentrations (*Figure 1D*). However, we observed a dose-dependent increase in FRET with low-nanomolar GR10 and PR10, indicating these DPRs are capable of binding to importin β and displacing the sensor. To further validate this observation, we used GFP nanobody-coated beads to bind Rango and probe for co-immunoprecipitation of importin β in the presence of increasing concentrations of DPRs (*Figure 1E–F*). We observed the dose-dependent displacement of importin β from the sensor at low nanomolar concentrations of GR10 and PR10, with little if any effect of GP10, GA10, or PA10, confirming that Rango release is responsible for the increases in FRET.

To test the functional consequence of R-DPR-importin β interactions for nuclear import, we performed the permeabilized cell assay (*Adam et al., 1990*), in which the plasma membrane of cultured cells is selectively permeabilized, leaving the nuclear membrane intact as verified by nuclear exclusion of 70 kD dextran (*Figure 1G*). Fluorescent transport cargo is then added, with energy regeneration mix and cell lysate to provide a source of importins and Ran for nuclear import, which is measured by increasing nuclear fluorescence. Traditionally, this method uses digitonin for permeabilization; however, when attempted with primary mouse cortical neurons, we found that even minimal concentrations of digitonin opened both the plasma and nuclear membranes. Since the nuclear envelope is devoid of the digitonin target cholesterol (*Colbeau et al., 1971*; *Adam et al., 1990*), we reasoned that its rupture in permeabilized neuronal cells was caused by mechanical perturbation upon depletion of cytoplasmic proteins. Therefore, we developed a protocol of rapid hypotonic cell opening in the presence of a concentrated BSA cushion, which protected the nucleus against rupture and facilitated the selective plasma membrane opening of neurons (*Figure 1—figure supplement 1*).

Using this method, we performed live imaging of nuclear import of Rango, a direct importin β cargo whose Ran-, importin β-, and energy-dependent nuclear translocation is conferred by the IBB domain (*Kaláb et al., 2006*). We verified that Rango import in permeabilized neurons is indeed dependent on energy and cell lysate, and can be inhibited by the importin β small molecule inhibitor, importazole (IPZ; *Soderholm et al., 2011*) in primary cortical neurons (*Figure 1—figure supplement 1*). Time-lapse imaging of Rango import for 30 min in permeabilized neurons showed no effect of GP10, GA10, or PA10 at up to 100 µM, whereas GR10 and PR10 showed dose-dependent inhibition of transport (*Figure 1H–J*, *Figure 1—figure supplement 1*). The reaction was allowed to reach steady-state and fixed at 2 hr, at which point we observed statistically significant transport inhibition beginning at 25 µM for both GR and PR (*Figure 1K*), with estimated IC50s as shown in *Figure 1L*. In contrast, only trace inhibition by GA10 and PA10 was seen even at 100 µM, and there was no effect of 100 µM GP10. To facilitate testing of a broader range of cargoes and concentrations, we performed the assay in HeLa cells, with similar results to those seen in neurons (*Figure 1L* and *Figure 1—figure supplement 2*). To verify that the behavior of Rango in nuclear import signals indeed corresponds to endogenous importin α/β complexes, we tested the effect of DPRs on import of GST-GFP-NLS (hereafter referred to as GFP-NLS), a similarly sized cargo that is loaded on importin β-bound importin α. Consistent with the expected lower efficiency of tripartite nuclear import complex assembly, R-DPRs perturbed GFP-NLS import even more strongly than that of Rango (*Figure 1L* and *Figure 1—figure supplement 2*).

The mechanisms of cargo recognition for importin β differ even from its structurally closest relative transportin, whose cargos are marked by the PY-NLS motif (*Lee et al., 2006*). However, since

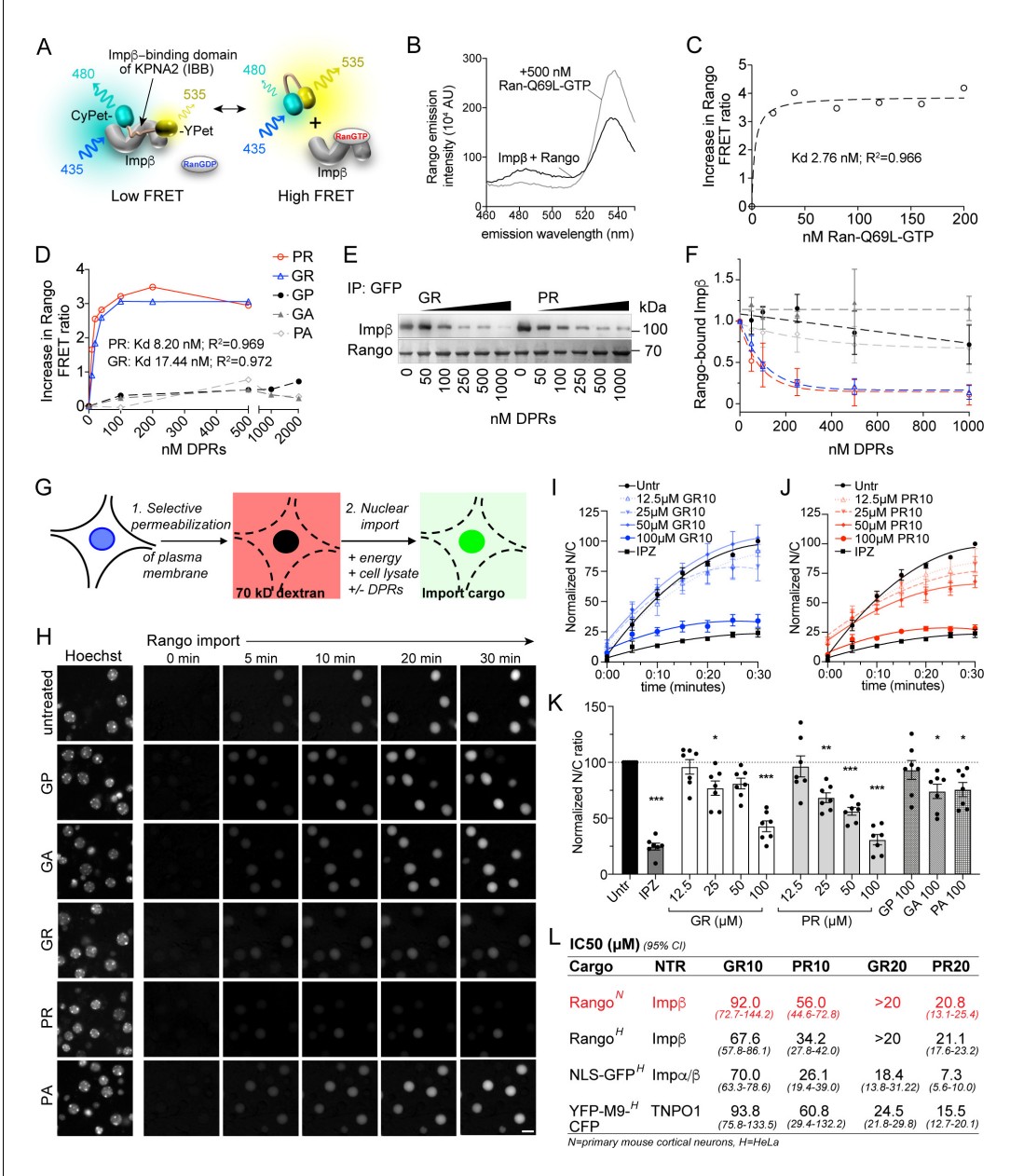

**Figure 1.** R-DPRs bind importin β and inhibit nuclear import. (**A**) Schematic of Rango FRET sensor, consisting of the importin β-binding domain (IBB) of importin α1 (KPNA2), flanked by CyPET (donor) and YPet (acceptor). (**B–C**) Rango spectral profile (**B**) and FRET ratio (**C**) demonstrating increase in FRET by adding hydrolysis-deficient Ran-Q69L-GTP to importin β-bound Rango (representative of three experiments). (**D**) Change in Rango FRET ratio induced by adding DPRs (10-mers) to importin β-bound Rango (representative of five experiments, data in C-D fit to non-linear model with one binding site). (**E**) GFP-trap co-immunoprecipitation of importin β by Rango in the presence of GR10 and PR10. (**F**) Quantification of Rango-bound importin β in (**E**), normalized to Rango and expressed as a fraction of untreated lysate (mean ± SD, three technical replicates, legend as in D). (**G**) Diagram of permeabilized cell nuclear import assay, which was adapted and validated for primary neurons (*Figure 1—figure supplement 1*). (**H**) Longitudinal wide-field images of Rango import in permeabilized mouse primary cortical neurons. Scale bar = 10 μm. (**I–J**) Nuclear to cytoplasmic (N/C) ratio of Rango import in (**H**), calculated by automated high content analysis. GR and PR graphs are separated for clarity; the control values are identical. All data are normalized to cells lacking energy/lysate and expressed as percent untreated controls (mean ± SEM of n = 4 biological replicates, 189 ± 125 cells per data point). (**K**) Steady state N/C ratio of Rango in primary neurons fixed after 2 hr (mean ± SEM of n = 7 biological replicates, 409 ± 202 cells per data point, *p<0.05, **p<0.01, ****p<0.001 vs. untreated cells, one-way ANOVA with Dunnett's post hoc test). (**L**) IC50 of R-DPRs for inhibition of

*Figure 1 continued on next page*

*Figure 1 continued*

nuclear import of designated cargoes, from (K) and *Figure 1—figure supplement 2*. 95% confidence intervals are shown (n = 3–6 biological replicates/condition, 409 ± 202 cells/ replicate for neurons, 1290 ± 305 cells/replicate for HeLa).>20 denotes conditions for which the IC50 was not reached up to the highest dose tested (20 µM). See source file for raw data and exact p values.

The online version of this article includes the following source data and figure supplement(s) for figure 1:

**Source data 1.** Raw data and p values for data in *Figure 1* and supplements.
**Figure supplement 1.** Neuron permeabilized cell assay validation.
**Figure supplement 2.** Extended HeLa nuclear import data.

---

the sequence of the PY-NLS also contains basic residues, we tested the effect of R-DPRs on the nuclear import of YFP-M9-CFP (hereafter referred to as M9), a transportin substrate based on the prototypic PY-NLS sequence of hnRNPA1 (*Siomi and Dreyfuss, 1995*; *Figure 1L* and *Figure 1—figure supplement 2*). Similar to the importin cargoes, M9 nuclear import in permeabilized HeLa cells was selectively inhibited by R-DPRs, which was more potent for PR than GR and approximately threefold more potent on average for 20mers than 10mers. Taken together, these results confirm that R-DPRs inhibit importin β-, importin α/β- and transportin-mediated nuclear import in this in vitro model system.

## R-DPRs interact with importin β in the bead halo assay

To further validate the direct interaction between R-DPRs and importin β, we performed the bead halo assay. This equilibrium-based binding assay is capable of identifying both low- and high-affinity interactions between 'bait' proteins immobilized to beads, and fluorescent 'prey' in the surrounding buffer, which forms a fluorescent halo on the bead surface (*Patel et al., 2007*; *Patel and Rexach, 2008*). First, we examined the propensity for all five DPRs to interact with biotinylated importin β, immobilized on the surface of neutravidin beads (*Figure 2A*). To quantify non-specific binding, we also tested bare beads and beads coated with biotinylated BSA. Confirming the specificity of the assay, the Rango sensor exclusively bound to importin β-coated beads, and neither of the control beads. Fluorescent dextran did not form a halo in any conditions. AF488-labeled PR10 and GR10 (200 nM) both showed modest non-specific binding to bare and BSA-coated beads, despite adjustments to pH, salt concentration, and the addition of detergent and BSA to minimize non-specific interactions. However, the R-DPR halo around importin β-coated beads was approximately two-fold more intense than controls (*Figure 2B*), as quantified by the ratio of the fluorescent rim of the beads (the intensity around the surface of the beads at their equator) to the background fluorescence (*Figure 2—figure supplement 1*). When 1 mg/ml neuronal lysate was added to test the stringency of the interaction, all binding between GR10 and the beads, including importin β, was lost (*Figure 2C–D*). For PR10, nonspecific binding decreased, but the intensity of the importin β halo persisted (and even slightly increased, perhaps due to recruitment of additional importin β- and PR-binding partners from the lysate). To further support the specificity of the observed interactions, we also tested the ability of free (unlabeled) importin β to compete for R-DPR binding to the importin β-coated beads, and saw that the haloes could be readily dispersed in a concentration-dependent manner (*Figure 2—figure supplement 2*). These findings further support a direct interaction between R-DPRs and importin β, while indicating a higher relative selectivity of PR for importin β, compared to GR.

## R-DPRs do not block passive nuclear influx

To test if the disruption of nuclear import resulted from changes in the passive exclusion limit of NPCs, we tested the effects of R-DPRs on the passive influx of small cargoes. Passive diffusion of GFP and small fluorescent dextrans into the nuclei of permeabilized HeLa cells was imaged at 10 s intervals for 5 min, and nuclear fluorescence quantified over time. All experiments were done in the context of energy and cell lysate, identical to the active transport conditions, so as not to miss putative effects that may depend on simultaneous active transport (i.e. recruitment of importins and DPRs to the NPC). Under these conditions, we observed the expected differences in the rates of passive influx of 10-, 40-, and 70-kD dextrans, and verified that addition of energy and lysate did not

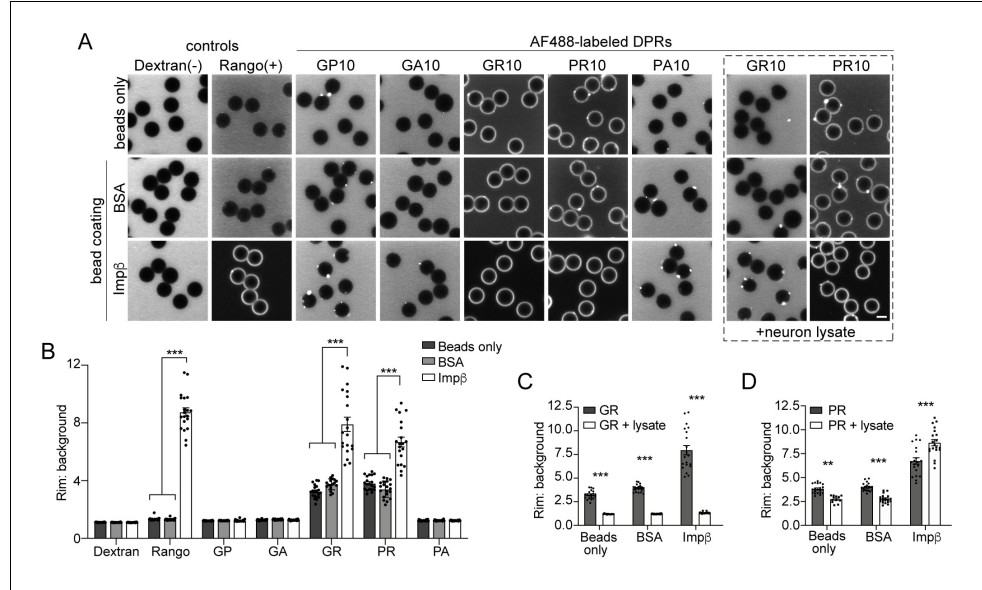

**Figure 2.** R-DPRs bind importin β in the bead halo assay. (**A**) Confocal images of AF488-labeled *C9orf72* DPRs added to neutravidin beads coated with biotinylated 'bait' proteins, in binding buffer or in the presence of 1 mg/ml neuron lysate (at right). FITC-dextran = negative control (-), Rango sensor = positive control (+). Scale bar = 4 μm. **B** Rim vs. background ratio in binding buffer (see *Figure 2—figure supplement 1* for quantification method). (**C–D**) Rim vs. background ratio for GR10 (**C**) and PR10 (**D**) in 1 mg/ml neuron lysate. In **B–D**, mean ± SEM is shown for n = 20 beads (5 intensity profiles/bead). **p<0.01, ***p<0.001 vs. control beads by two-way ANOVA with Tukey post-hoc test. See source file for raw data and exact p values.

The online version of this article includes the following source data and figure supplement(s) for figure 2:

**Source data 1.** Raw data and p values for data in *Figure 2* and supplements.

**Figure supplement 1.** Quantification method for bead halo assay.

**Figure supplement 2.** Free importin β competes for R-DPR binding to importin β-coated beads.

---

affect the baseline rate of passive influx of GFP (27 kD, no NLS) (*Figure 3—figure supplement 1*). When we preincubated permeabilized nuclei with high concentrations of R-DPR 10mers or 20mers for 30–60 min, we observed no slowing of passive nuclear influx (*Figure 3*). Instead, R-DPRs caused apparent acceleration of the nuclear influx of both GFP and 40-kD dextran, an effect that was specific to R-DPRs and not observed for GP10, GA10, or PA10 (*Figure 3—figure supplement 1*).

The rate of passive transport through the NPC is governed by the FG-Nup barrier (*Mohr et al., 2009*; *Timney et al., 2016*; *Frey et al., 2018*), in addition to local concentrations of importin β and RanGTP (*Ma et al., 2012*; *Kapinos et al., 2017*). R-DPR/FG-domain interactions are predicted based on the propensity of arginines to undergo cation-pi interactions with aromatic phenylalanine rings (reviewed by *Gallivan and Dougherty, 1999*; *Banani et al., 2017*). Indeed, previous interactome studies predicted R-DPR interactions with FG-Nups including Nup54, 62, 98, 153, and 214 (*Lin et al., 2016*; *Yin et al., 2017*), and PR20 peptides were shown to bind and stabilize Nup54 and 98 FG-domain polymers, a property hypothesized to decrease NPC permeability for passive and active transport (*Shi et al., 2017*). To test for direct DPR/FG binding, we probed for interactions between the five *C9orf72* DPRs and FG-domains of yeast homologs of Nup62 (Nsp1) and Nup98 (Nup100 and Nup116) in the bead halo assay (*Figure 3—figure supplement 2*). These represent two distinct categories of FG-motifs, the more charged FxFG of the Nsp1 C-terminal region, versus the more hydrophobic and less-charged GLFG-domains of Nup100 and Nup116, the latter of which are particularly highly conserved and thought to be critical for determining the mesh-like properties of the FG hydrogel (reviewed by *Schmidt and Görlich, 2016*).

We again observed non-selective binding by the R-DPRs, including to beads coated with a Nup116 F->A mutant construct. However, quantification of the halo intensities showed modest selective binding of GR10 and PR10 to GLFG-domains of Nup100, but not Nsp1 or Nup116, in a

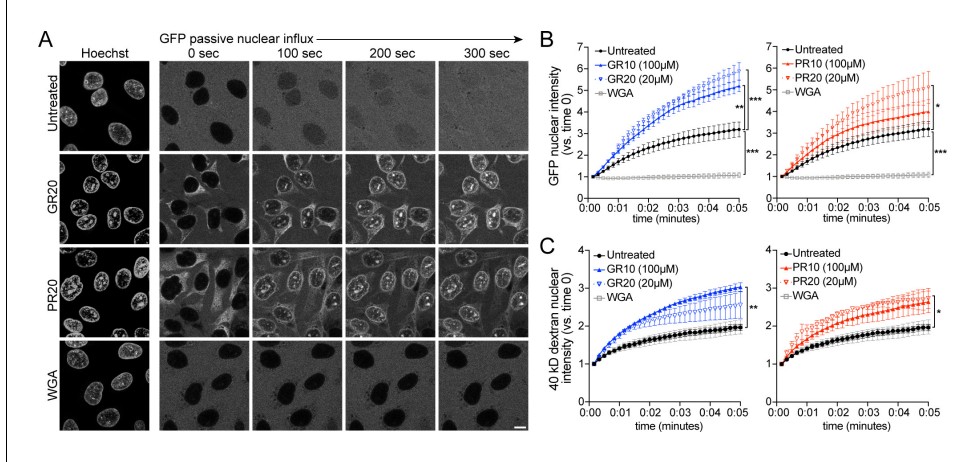

**Figure 3.** PR and GR accelerate passive nuclear influx. (**A**) Confocal time-lapse imaging of GFP nuclear influx in permeabilized HeLa cells following ≥ 30 min. incubation with buffer (untreated), 20 μM GR20, 20 μM PR20, or 0.8 mg/ml wheat germ agglutinin (WGA, positive control). Scale bar = 10 μm. (**B–C**) Nuclear GFP (**B**) and 40 kD dextran (**C**) intensity normalized to background fluorescence, expressed vs. time 0 (no influx = 1). GR and PR are separated for clarity; the control values are identical. All experiments included lysate and energy. See *Figure 3— figure supplement 1* for validation of assay conditions and non-R-DPR testing, and *Figure 3—figure supplement 2* for binding studies with FG-domains which contribute to the NPC selectivity barrier. Data are mean ± SEM for n = 3–6 biological replicates/condition (20–30 cells/replicate). *p<0.05, **p<0.01, ***p<0.001 vs. untreated cells at 5 min by one-way ANOVA with Dunnett's post hoc test. See source file for raw data and exact p values.

The online version of this article includes the following source data and figure supplement(s) for figure 3:

**Source data 1.** Raw data and p values for data in *Figure 3* and supplements.
**Figure supplement 1.** Validation of passive nuclear influx assay.
**Figure supplement 2.** R-DPRs show modest binding to FG-domains in the bead halo assay, which can be augmented by importin β.

---

pattern that appeared primarily dependent on the length of the FG-domains tested (Nup100$_{1-607}$ > Nup100$_{1-310}$>Nsp1$_{497-609}$>Nup116$_{348-458}$). For PR10, FG-binding could be augmented (to Nup100 and Nsp1 fragments) by adding unlabeled importin β to the assay, suggesting that recruitment of PR to FG-domains at the NPC could be mediated in part by an indirect interaction through importin β. Overall, these results support modest direct and indirect binding of R-DPRs to FG domains, which based on our passive transport data, do not decrease NPC permeability and even increase passive influx, by a mechanism that remains to be elucidated.

## R-DPR-induced aggregates recruit NCT proteins

Upon addition of R-DPRs to cell lysate for the transport assays, we observed the rapid formation of insoluble aggregates (*Figure 4A*). To identify the components of these aggregates and determine their potential relevance for the nuclear import defect, we spun them down and analyzed their protein content via mass spectrometry (*Figure 4A–B*; data available via ProteomeXchange with identifier PXD015656). 858 proteins were identified in each of two GR replicates and 758 in two PR replicates, with 647 (67%) in common. Consistent with previous reports, these included numerous nucleic acid-binding proteins and ribosomal subunits. Gene ontology (GO) analysis confirmed enrichment of nucleolar proteins, ribonucleoproteins, spliceosomal complex subunits, stress granule constituents, and others (*Figure 4—figure supplement 1*). Among these, low complexity domain (LCD)-containing proteins implicated in ALS/FTD were identified including TDP-43, FUS, Matrin-3, and hnRNPs. Multiple NCT proteins including karyopherins, Nups, Ran cycle proteins, and THO complex proteins, which participate in mRNA biogenesis and nuclear export (*Rondón et al., 2010*), were also found among the identified targets (*Figure 4A–B*).

Next, we validated a subset of these identified proteins by an immunoblotting-based sedimentation assay. We added each of the five DPRs to transport cell lysate in the presence of energy, allowed aggregates to form for 1 hr, and pelleted the aggregates by centrifugation. Supernatants

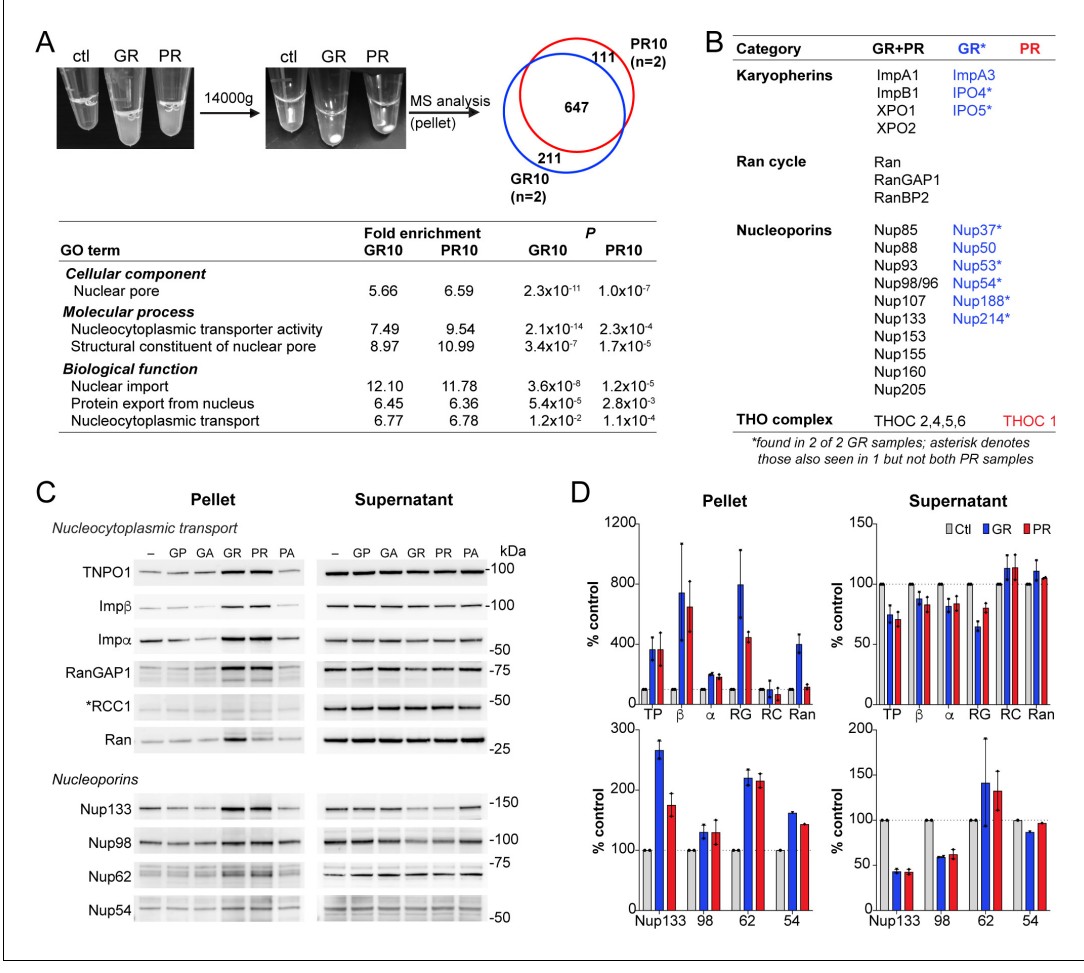

**Figure 4.** R-DPR-induced aggregates recruit NCT proteins. (A) Aggregates formed by adding R-DPRs to HEK cell lysate in transport buffer (before and after 15 min centrifugation). Venn diagram indicates number of proteins identified by mass spectrometry analysis of pellets (n = 2 technical replicates). Enriched NCT-related GO terms are shown, with fold change and p value calculated by the DAVID algorithm. Overall top GO terms are shown in *Figure 4—figure supplement 1*. (B) List of identified NCT-related proteins, in all 4 samples (black), n = 2 GR10 samples (blue), and n = 2 PR10 samples (red). Asterisk denotes samples seen in n = 2 GR10 samples and only n = 1 PR10 sample. (C) Western blots for indicated NCT and Nup proteins in pellet vs. supernatant fractions. RCC1 is marked with an asterisk, as this protein was not identified in the MS results and serves as the negative control. All samples were loaded by volume, see *Figure 4—figure supplement 2* for membrane protein stain and additional Western blots of disordered RNA binding proteins. (D) Quantification of blots in (C). Mean ± SD for two technical replicates is shown (TP = TNPO1, β = importin β, α = importin α, RG = RanGAP1, RC = RCC1, Ran = RanGTPase). See source file for raw data.

The online version of this article includes the following source data and figure supplement(s) for figure 4:

**Source data 1.** Raw data and p values for data in *Figure 4* and supplements.
**Figure supplement 1.** Overall top GO terms enriched in R-DPR aggregates.
**Figure supplement 2.** Western blots for selected low complexity-domain (LCD)-containing proteins in R-DPR supernatant vs. pellet fractions.

were removed, and the pellets rinsed with PBS before preparing samples of supernatants and pellets for separation by SDS-PAGE. Samples corresponding to equal volumes of supernatants and pellets (precipitated from equal fractions of the initial extract), were then analyzed by immunoblotting. In this manner, we assessed the the partitioning of NCT proteins, Nups, and LCD-containing proteins between the soluble and aggregate fraction of the extracts (*Figure 4C–D* and *Figure 4—figure supplement 2*). We saw R-DPR-mediated enrichment in the pellet for importin β, RanGAP1, transportin-1, Ran, and importin α, with only minor decreases in the supernatant. RCC1 was not identified by

mass spectrometry, and as predicted did not sediment with the DPRs, serving as a negative control. We also confirmed deposition of Nups 54, 62, 98, and 133 in the pellet (*Figure 4C–D*), along with the low complexity domain (LCD)-containing RNA binding proteins TDP-43, FUS, Matrin-3, hnRNPA1, and hnRNPA2/B1, ribosomal protein RPS6, and the ATP-dependent RNA helicase DDX3X (*Figure 4—figure supplement 2*). As opposed to the NCT proteins, many of these LCD-containing proteins were markedly or completely depleted from the supernatant.

These data confirm that R-DPR aggregates can recruit NCT constituents in addition to a host of nucleic acid-binding proteins. However, NCT proteins were not substantially depleted from the supernatant even in the presence of 100 µM GR10 and PR10, suggesting that sequestration of critical NCT factors in these insoluble protein assemblies is unlikely to fully explain the failure of nuclear import in the transport assays.

## R-DPR nuclear import blockade does not require aggregates and is rescued by RNA

Cytoplasmic aggregate formation, a pathological hallmark of neurodegenerative disease, has been proposed as a general mechanism for impairment of NCT by sequestration of critical NPC and NCT proteins (*Woerner et al., 2016*). However, there is no evidence to date that such accumulation alters or disorganizes the NPC, and remains to be demonstrated whether it is the process of aggregate formation and sequestration, or the disordered proteins themselves, that disrupt NCT. To address this question in the context of R-DPR aggregates, we tested several approaches for preventing aggregate formation in our model system. Addition of the aliphatic alcohol, 1,6-hexanediol, previously shown to disrupt GR- and PR-induced protein assemblies (*Lee et al., 2016*), was incompatible with transport and caused dose-dependent inhibition at baseline (*Figure 5—figure supplement 1*). This is likely due to disruption of FG-domains within the central channel, as previously reported (*Ribbeck and Görlich, 2002*). NTRs themselves, as hydrophobic interactors of aggregation-prone RNA binding proteins, have been shown to promote solubility of their cargoes and may have evolved in part as cytoplasmic chaperones (*Jäkel et al., 2002*; *Guo et al., 2018*; *Hofweber et al., 2018*; *Yoshizawa et al., 2018*; *Qamar et al., 2018*). However, even low concentrations of exogenous, full-length importin β inhibited nuclear import when added to the transport assay, likely due to sequestration of Rango and available RanGTP. Moreover, neither 1,6-hexanediol nor exogenous importin β could reverse mild nuclear import inhibition due to 25 µM PR10 (*Figure 5—figure supplement 1*).

Next, we tested the effect of increasing the concentration of RNA, based on the growing evidence that RNA mediates the solubility of intrinsically disordered proteins, including FUS, TDP-43, and RNA-dependent DEAD-box ATPases (*Maharana et al., 2018*; *Hondele et al., 2019*), and attenuates TDP-43 inclusions and neurotoxicity in vitro (*Mann et al., 2019*). PolyU RNA has also been shown to colocalize with PR20 in phase separated droplets (*Boeynaems et al., 2017*). We hypothesized that RNA may interact with R-DPRs and perhaps change the material properties of R-DPR induced aggregates in the transport assay. When total HEK cell RNA was added to the transport reaction, we indeed saw a dose-dependent rescue of nuclear import that was RNAse-sensitive (*Figure 5A*). This did not appear to be attributable to a significant reduction of the quantity of insoluble material in the transport reaction, either by protein stain or Western blot (*Figure 5—figure supplement 2*), although these studies were not aimed to detect changes in aggregate size or composition. Electrophoretic mobility shift assays confirmed that in a purified system, RNA of a broad range of sizes binds to R-DPRs (*Figure 5—figure supplement 3*).

Since the above approaches for mitigating aggregate formation were either ineffective or incompatible with the transport assay, we modified the assay to clarify the role of R-DPR induced aggregates in the disruption of nuclear import. Before adding the R-DPR-treated cell lysates to permeabilized cells, we separated the soluble phase of the extracts (supernatants) from the aggregates (pellets) by centrifugation (diagrammed in *Figure 5B*). We reasoned that if aggregates functionally sequester critical transport factors, the remaining supernatant would be insufficient to drive nuclear import. However, if the aggregates contain inhibitor(s) of nuclear import or are themselves inhibitory, depleting them could rescue transport impairment. The results were markedly different for GR versus PR (*Figure 5C*). For GR10, removing the insoluble pellet restored nuclear import to normal, confirming that the inhibitory factor was present in (or was) the aggregates. In contrast,

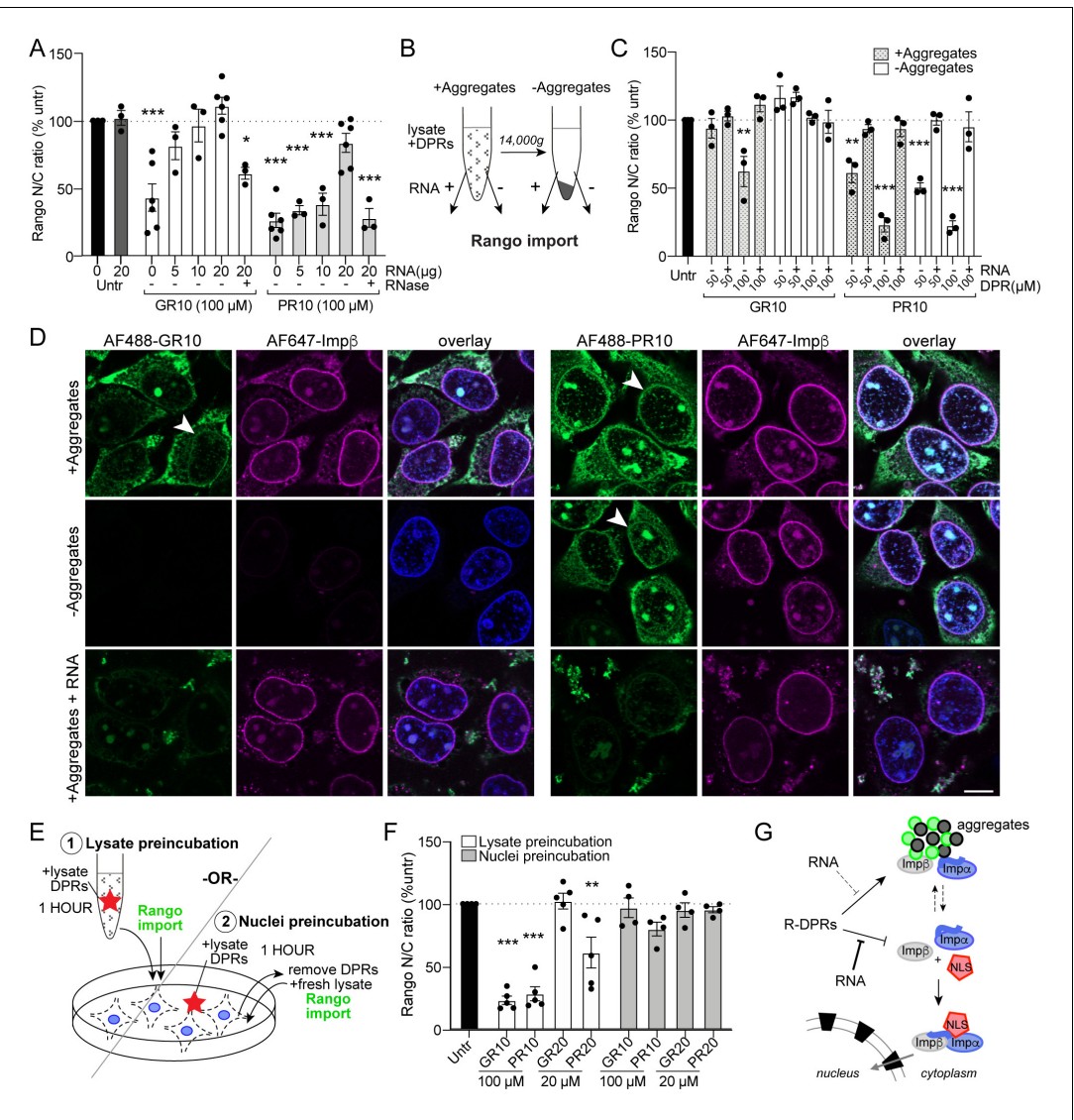

**Figure 5.** R-DPR nuclear import blockade does not require aggregates and is rescued by RNA. (**A**) Rango N/C ratio in permeabilized HeLa transport reactions with 100 μM GR10 or PR10 and increasing concentrations of total HEK cell RNA +/- RNAse. See *Figure 5—figure supplement 1* for attempts to rescue with 1,6-hexanediol and importin β. (**B**) Schematic of fractionated Rango transport assays, run with aggregates present or absent (supernatant only), followed by addition of RNA to a subset of reactions. See *Figure 5—figure supplement 2* for western blots of fractionated samples ± RNA. (**C**) Rango N/C ratio from fractionated transport assays. (**D**) Confocal images of fractionated transport assays run in the presence of AF488-labeled R-DPRs and AF647-labeled importin β. Arrows mark R-DPR collection around the nuclear membrane in conditions where transport was inhibited. Acquisition parameters were kept constant for all images (scale bar = 10 μm). (**E**) Schematic of (1) lysate vs. (2) nuclei R-DPR preincubation assays. (**F**) Rango N/C ratio from preincubation assays. (**G**) Working model: R-DPRs block nuclear import by binding to importin β and preventing the formation of the importin α•importin β•NLS cargo complex in the soluble phase of the transport reaction, which can be alleviated by RNA. See *Figure 5— figure supplement 3* for DPR/RNA electrophoretic mobility shift assay. For A,C,F mean ± SEM of n ≥ 3 biological replicates are shown (each data point represents 1462 ± 555 cells). *p<0.05, **p<0.01, ***p<0.001 vs. untreated cells by one-way ANOVA with Dunnett's post-hoc test. See source file for raw data and exact p values.

The online version of this article includes the following source data and figure supplement(s) for figure 5:

**Source data 1.** Raw data and p values for data in *Figure 5* and supplements.
**Figure supplement 1.** 1,6-HD and importin β do not rescue nuclear import in the permeabilized cell assay.
**Figure supplement 2.** RNA only minimally attenuates R-DPR aggregate formation.
**Figure supplement 3.** RNA binds R-DPRs in an electrophoretic mobility shift assay.

nuclear import remained perturbed in the supernatants of the PR10 aggregates, and was restored by the addition of RNA.

Next, we monitored the location of the R-DPRs with respect to the aggregates and the NPC by adding AF488-labeled DPRs and AF647-labeled importin β to the transport reactions. By confocal microscopy, we observed that the transport disruption correlated with the presence of GR and PR in the vicinity of the nuclear envelope (*Figure 5D*). AF488-GR10 fully sedimented into the pellet, leaving no visible GR10 in the supernatant, where transport proceeded normally. In contrast, a subset of AF488-PR10 remained in the supernatant and was present at the nuclear envelope, paralleling the persistent inhibition of nuclear import by the PR10 supernatants. RNA dispersed AF488-R-DPRs from the permeabilized cell nuclei in all conditions, restoring nuclear import. These results suggest that the import inhibition depends on GR or PR acting directly, rather than through putative intermediary factor(s), to inhibit nuclear import. The strikingly divergent segregation of GR vs. PR between the supernatant and pellet demonstrates that, while both bind importin β in a purified system, in the context of cell lysate, PR more readily dissociates from the aggregates into the soluble phase, where it can bind and inhibit importin β.

The critical steps of importin β-mediated nuclear import take place at NPCs via interactions with FG-Nups. To test whether the interaction between R-DPRs and the NPC is sufficient to block import, we ran two parallel sets of import reactions (diagrammed in *Figure 5E*). In the 'lysate preincubation' paradigm, as for previous active import assays, R-DPRs were added to lysates used to supply transport factors, preincubated for 1 hr, and then added to permeabilized cells along with Rango and energy to initiate the transport reaction. In the 'nuclei preincubation' set, we first exposed the permeabilized cell nuclei to R-DPRs (in the presence of lysate and energy, but no fluorescent cargo). After 1 hr, the DPR-lysate mix was removed from the nuclei, and fresh transport lysate, energy, and cargo added to initiate transport (without R-DPRs). We hypothesized that, if the R-DPRs inhibited Rango import by associating with and perturbing the NPC, we should see reduced import in the 'nuclei preincubation' group. However, transport proceeded normally (*Figure 5F*). These results support a model in which the R-DPRs inhibit nuclear import by directly interfering with factor(s) present in the soluble phase of the NCT machinery (*Figure 5G*), which is consistent with the hypothesis that DPRs interfere with the cargo loading on importins.

## Discussion

In this study, we investigated disruption of importin-mediated nuclear import by mutant *C9orf72*-associated R-DPRs. We observed that R-DPRs bind importin β at low-nanomolar concentrations, disrupt its interaction with the importin α IBB domain, and impair nuclear import of importin β and importin α/β cargos in permeabilized cells. Nuclear import by transportin, a related member of the importin class that recognizes the structurally distinct PY-NLS, was also disrupted. Import inhibition in the permeabilized cells could not be explained by aggregate-mediated sequestration of NCT machinery; rather, our data support a model in which R-DPRs, particularly PR, directly interfere with importin-cargo loading.

### Molecular mechnanisms of R-DPR nuclear import inhibition

The high arginine content of R-DPRs predicts several molecular mechanisms through which they could interfere with importin•cargo complex formation. Since positively charged arginine and lysine residues are characteristic of several classes of NLSs, R-DPRs could compete with or displace such cargoes from importins by charge-based mimicry. In addition to the IBB domains (discussed below), such K/R-rich NLSs include the 'classic' mono- and bipartite NLS for importin α, the R-rich NLS of ribosomal proteins that directly bind importin β, importin 5, and importin 7 (*Jäkel, 1998*), the basic N-terminal portion of the transportin PY-NLS, and the SR-NLS of transportin 3 (*Soniat and Chook, 2015*). The potential for IBB domains as targets of R-DPR mimicry is evidenced both by the presence of K/R-rich regions and their relevance for a wide variety of cargos carried by the seven importin α isoforms in humans, and the otherwise structurally unrelated snurportin 1, which also contains an IBB domain (*Lott and Cingolani, 2011*; *Oka and Yoneda, 2018*). Thus, R-DPR cargo mimicry could potentially slow the loading of NLS cargos on many diverse types of importins in parallel. Second, electrostatic cation-pi interactions of arginines with aromatic rings (*Gallivan and Dougherty, 1999*; *Banani et al., 2017*) could attract R-DPRs to the tyrosine of the transportin PY-NLS, disrupting in

this case the cargo, rather than the karyopherin. Notably, consensus nuclear export signals (NES) recognized by exportins do not depend on K/R-rich motifs, but rather three dimensional structures based on hydrophobic residues, including aromatic phenylalanines (*Xu et al., 2012*). Thus, although not examined here, exportin-NES interactions could also be be subject to electrostatic interference by R-DPRs. Future studies will be needed to examine exactly how R-DPRs target each class of karyopherins, and the hierarchy of the relevance of such interactions to the mechanisms of R-DPR cellular toxicity.

## Lack of R-DPR blockade of NPC channels

Previously published interactomes (*Lin et al., 2016*; *Yin et al., 2017*) and mechanistic studies (*Shi et al., 2017*) predict binding between R-DPRs and FG-Nups, which line the central channel of the NPC and are fundamental for establishing the permeability barrier (reviewed by *Schmidt and Görlich, 2016*). *Shi et al. (2017)* proposed that stabilization of FG-Nup polymers and the ensuing blockade of NPC passage may be responsible for PR20-mediated inhibition of nuclear import. To explore this further, we tested the effect of R-DPR 10- and 20-mers on passive nuclear influx. Surprisingly, we observed acceleration rather than slowing. The precise cause is unclear. Studies in yeast suggest that to pass through the NPC, cargos must overcome collisions with highly mobile, disordered FG domains, particularly the Nup98 homologues Nup100 and Nup116 (*Timney et al., 2016*). If R-DPRs directly or indirectly target these FG domains, as suggested by Shi et al., and by our bead halo studies, this could drastically increase their positive charge or reduce their mobility and thus perturb their gate-keeping function. In addition to FG-Nups, the passive properties of the pore have also been shown to depend on karyopherins. Using superresolution microscopy, *Ma et al. (2012)* showed that when extra importin β accumulates within the NPCs, concentrating inside the peripherally-localized active transport zone, the central passive transport channel widens. Alternatively, by monitoring the passive cargo transport rate, *Kapinos et al. (2017)* showed that when importin•cargo complexes were depleted from the NPC, the pore became leaky. Permeabilized cell experiments in which we monitored the location of AF647-labeled importin β (*Figure 5D*) suggest that in the presence of the R-DPR-transport blockade, importin β indeed accumulates at the nuclear rim. Superresolution microscopy would be needed to show the precise site of importin β accumulation in this paradigm, to further clarify if either of these proposed mechanisms may contribute. Nevertheless, our passive transport data (*Figure 3*) do not support the conclusion that R-DPRs induce a blockade to nuclear transport by occluding the NPC. This conclusion is also strongly supported by the results of the 'nuclei preincubation' experiment (*Figure 5E–F*), in which no inhibition of active import was observed despite prolonged exposure of permeabilized nuclei to R-DPRs.

## Divergent properties of GR and PR

Throughout this study, we observed marked differences in the behavior of GR and PR. Although both readily displaced importin β from the Rango FRET sensor, PR showed greater potency for inhibiting nuclear import in the permeabilized cell assay and higher relative selectivity for importin β in the bead halo assays. Moreover, while both PR and GR readily induced aggregate formation in the transport lysates, PR showed greater tendency to dissociate from the aggregates (*Figure 5D*). Based on their divergent molecular properties, the differing behavior of GR and PR is not surprising. On one hand, both are positively charged, with a predicted disordered state and polynucleotide binding activity (https://predictprotein.org/; *Yachdav et al., 2014*). However, because glycine is conformationally flexible and its side chain is comprised of a single hydrogen, while the integrated proline cyclic side-chain induces a rigid bend in the peptide backbone, GR is predicted to accommodate many different macromolecular ligands, while PR may be more selective. As recently reviewed, GR and PR have also been found to have differing half-lives, intracellular localization, and propensity to disrupt stress granule dynamics and mitochondrial function (*Freibaum and Taylor, 2017*). PR20, but not GR20, was also recently shown to inhibit proteosomal degradation (*Gupta et al., 2017*), and a recent comparison of modifiers of GR and PR toxicity in yeast showed little overlap (*Chai and Gitler, 2018*). Although our MS analysis did not permit quantitative comparison between PR and GR aggregates, 21% of hits were unique to GR, and 11% to PR, with varying selectivity for target proteins also observed by western blot. This lack of overlap was also reported

for GR50 and PR50 interactomes, which showed 35% and 25% unique hits, respectively (*Lee et al., 2016*).

Thus, although in a purified system GR and PR show a similar propensity to bind importin β, in the more complex environment of the cell, our findings predict PR to be the more potent disruptor. Micromolar concentrations of R-DPRs were required to observe functional import blockade in the permeabilized cell assay; however, 20mers were on average 3.3-fold more potent than 10mers across all cargoes, suggesting that longer DPRs, as are likely present in patients, may be significantly more potent. However, the size of polyGR and polyPR peptides in patients is unknown. GGGGCC repeat lengths in the 1000s have been reported in postmortem brain (*van Blitterswijk et al., 2013*; *Dols-Icardo et al., 2014*; *Nordin et al., 2015*), although the processivity of ribosomes along the repeat RNA, and what terminates repeat-associated non-AUG translation, is unclear. High-molecular-weight smears have been observed by SDS-PAGE (*Zu et al., 2013*). By ELISA, the poly-GP concentration in postmortem motor cortex was estimated at a median of 322 ng/mg protein (*Gendron et al., 2015*), but comparable measurements for R-DPRs, and in particular the relative abundance of GR vs. PR in patient tissue, remain to be determined.

## RNA-mediated rescue of the R-DPR import blockade

Based on growing evidence that RNA is integral to the solubility of disordered protein assemblies (*Maharana et al., 2018*; *Langdon et al., 2018*; *Hondele et al., 2019*), and polyU RNA can phase separate with PR (*Boeynaems et al., 2017*), we tested the effect of adding total cellular RNA to the transport reaction, and observed dose-dependent rescue. Total protein aggregates were not substantially reduced by the RNA, based on our sedimentation assays; however, significantly less AF488-labeled R-DPRs were observed in the vicinity of the nuclear envelope by confocal microscopy. Our electrophoretic mobility shift assay shows that a broad range of cellular RNAs can bind to R-DPRs directly, and previous evidence in a purified system showed that synthetic RNAs can facilitate suspension of R-DPRs in a droplet-like state (*Boeynaems et al., 2017*), indicating that direct sequestration of R-DPRs by RNA could contribute to the reduced deposition of AF488-DPRs along the nuclear envelope, and the beneficial effects on nuclear import. At the same time, RNA could act indirectly to sequester the R-DPRs away from importins, by reducing the average aggregate size and thus increasing the number of exposed R-DPR binding sites. While future studies will be needed to fully elucidate the mechanisms of direct and indirect effects of RNA on R-DPRs, our data suggest that, at least in the permeabilized cell model, RNA can mitigate aberrant protein-protein interactions in a functionally meaningful way.

## DPR-mediated NCT disruption in *C9orf72*-ALS

Initial reports of NCT disruption as a pathogenic mechanism in *C9orf72*-ALS were based on genetic modifier screens of expanded repeat RNA and DPRs in *Drosophila* and yeast. These studies implicated numerous karyopherins, Nups, and Ran cycle proteins as potential direct and indirect mediators of *C9orf72*-ALS pathogenesis (*Zhang et al., 2015*; *Jovičić et al., 2015*; *Freibaum et al., 2015*; *Boeynaems et al., 2016*). RanGAP1 was identified as a direct interactor of expanded repeat RNA, providing one mechanism by which mutant *C9orf72* repeat RNA may impair NCT, via disruption of the RanGTP gradient (*Zhang et al., 2015*).

Additional studies have since investigated mechanisms by which individual DPRs may disrupt the nuclear transport apparatus. Cytoplasmic expression of poly-GA, a hydrophobic DPR proposed to form amyloid fibrils (*Chang et al., 2016*), was shown to impair importin α/β- but not transportin-mediated import in transfected HeLa and primary hippocampal neurons (*Khosravi et al., 2016*). Overexpression of Nup54, Nup62, and importin α3 rescued GA-mediated import disruption, suggesting these factors may be rate-limiting due to sequestration within cytoplasmic aggregates. PR20 was previously shown by STED microscopy to localize to the central channel of the NPC in *Xenopus* oocytes (*Shi et al., 2017*), and impeded nuclear import of NLS-BSA in permeabilized HeLa cells. The import blockade was attributed to a propensity for PR20 to bind FG domains and stabilize them in a polymerized state, altering the permeability barrier of the NPC. While the current study was under review, Vanneste and colleagues reported that they were unable to impede nuclear transport of fluorescent NLS-NES shuttle proteins by addition of synthetic GR20 or PR20 to HeLa cells in the culture media, or lentiviral expression of mCherry-tagged poly-GR100 and poly-PR100 in multiple cell types

(*Vanneste et al., 2019*). Poly-GA100 also failed to inhibit nuclear import in HeLa or SH-SY5Y cells, and caused only a mild decrement in iPSC-derived motor neurons.

These examples highlight several mechanistic discrepancies. Based on its markedly different biophysical properties, it is not surprising that GA would behave differently with respect to the NCT apparatus than R-DPRs. The 'sequestration model', in which cytoplasmic aggregates consume NCT components, is also supported by recent studies of artificial β-sheet constructs (*Woerner et al., 2016*) suggesting that strongly hydrophobic proteins may share this behavior. While we did not see NCT inhibition by GA10 in permeabilized HeLa, and only minimal inhibition at high concentrations in permeabilized neurons, we also did not observe GA-induced aggregate formation. Longer peptides and/or prolonged incubation time may be needed to further test the sequestration capacity of poly-GA in the permeabilized cell model.

Our current study, and that of *Shi et al. (2017)*, used the permeabilized cell assay to study PR-mediated inhibition of nuclear import. Although we both observed dose-dependent inhibition of cargo import, our conclusions differ with respect to the mechanism of import disruption, which we argue occurs via karyopherin disruption, rather than the interaction with FG-Nups, for reasons discussed above. Less clear is the reason for the lack of effect reported by *Vanneste et al. (2019)*. We did observe promiscuous binding tendencies of GR in our assays, which could predict that GR rapidly becomes sequestered in living cells by other binding partners (i.e. ribosomes, histones, and other disordered proteins), preventing any significant disruption of the highly abundant karyopherins. PR expressed from tagged, randomized codon constructs is localized nearly exclusively in nuclei and concentrated in nucleoli (*Wen et al., 2014*; *Khosravi et al., 2016*; *Vanneste et al., 2019*), which is distinct from the localization seen in human postmortem issue, where cytoplasmic aggregates of PR are seen in addition to the nuclear signal (*Gendron et al., 2013*). It is possible that sequestration of PR in the nucleolus may protect against disruption of nuclear transport. Detailed investigation of endogenous nuclear transport receptors and cargoes in *C9orf72*-ALS patient tissue is needed, to help clarify the above discrepancies in an environment where RNA foci and all five DPRs are simultaneously expressed. Indeed, multiple gains of toxic function may converge in the pathologic cascade of this disease.

## Conclusion

In summary, we propose a model in which R-DPRs bind and interfere with importin•cargo loading at the NPC. Based on these findings, we speculate that importin β disruption may contribute to pathological protein mislocalization in *C9orf72*-mediated ALS/FTD, including TDP-43, for which links to downstream neurodegeneration are beginning to be unraveled (*Ling et al., 2015*; *Melamed et al., 2019*; *Klim et al., 2019*). Further investigation is needed regarding disruption of endogenous cargoes in *C9orf72* patient tissue, and the potential for use of RNA-based strategies to mitigate aberrant R-DPR protein-protein interactions.

## Materials and methods

**Key resources table**

| Reagent type (species) or resource | Designation | Source or reference | Identifiers | Additional information |
|---|---|---|---|---|
| Cell line (*Homo sapiens*) | HeLa | ATCC *Kaláb et al., 2006* | clone HeLa 61 | Single cell-derived clone |
| Cell line (*Homo sapiens*) | HEK293T | ATCC | | |
| Cell line (*M. musculus*) | Primary cortical neurons (embryonic) | This paper | | Harvested from timed pregnant C57BL/6J females |

*Continued on next page*

*Continued*

| Reagent type (species) or resource | Designation | Source or reference | Identifiers | Additional information |
|---|---|---|---|---|
| Recombinant DNA reagent | pRSET zzRanQ69L | This paper | pKW1234; pK1097 | *E. coli* expression of Protein A-tagged human RanQ69L |
| Recombinant DNA reagent | pRSET zzRCC1 | This paper | pKW1907; pK1098 | *E. coli* expression of Protein A-tagged human RCC1 |
| Recombinant DNA reagent | pRSET Rango-2/α1+linkers | This paper | pK44 | *E. coli* expression of Rango 2 with importin α1 IBB, optimized for FRET |
| Recombinant DNA reagent | pRSET Rango-2/α1 | This paper | pK188 | *E. coli* expression of Rango 2 with importin α1 IBB |
| Recombinant DNA reagent | pRSET GFP-AviTag | This paper | pK803 | *E. coli* expression of GFP, with C-terminal Avitag, used in non-biotinylated form here |
| Recombinant DNA reagent | pRSET Importin β-AviTag | This paper | pKW1982; pK1099 | *E. coli* expression of WT human Importin β, with C-terminal biotin tag |
| Recombinant DNA reagent | BirA ligase | Avidity.com | AVB101 | *E. coli* expression of untagged BirA biotin ligase for biotinylation of co-expressed AviTag proteins |
| Recombinant DNA reagent | pRSET YFP-M9-CFP | *Soderholm et al., 2011* | pKW1006 | *E. coli* expression of fluorescent M9, TNPO1 cargo |
| Recombinant DNA reagent | pET30a 6His-S-Importin β$_{(1-876)}$ | *Chi et al., 1997* | pKW485 | *E. coli* expression of WT S-tagged human Importin β |
| Recombinant DNA reagent | pGEX GST-GFP-NLS | *Levy and Heald, 2010* | pMD49 | *E. coli* expression of fluorescent importin α cargo with SV40 NLS |
| Recombinant DNA reagent | pGEX-2TK-Nup100$_{(1-610)}$ | *Onischenko et al., 2017* | pKW2960, ID 370 | *E. coli* expression of truncated GST-tagged GLFG domain of yeast Nup100 (human Nup98 homologue |
| Recombinant DNA reagent | pGEX-2TK-Nup100$_{(1-307)}$ | *Onischenko et al., 2017* | pKW2959, ID369 | *E. coli* expression of GST-tagged GLFG domain of yeast Nup100 (human Nup98 homologue) |
| Recombinant DNA reagent | pGEX-2TK-Nup116$_{(348-458)}$ | *Onischenko et al., 2017* | pKW2907, ID350 | *E. coli* expression of GST-tagged GLFG domain of yeast Nup116 (human Nup98 homologue) |

*Continued*

| Reagent type (species) or resource | Designation | Source or reference | Identifiers | Additional information |
|---|---|---|---|---|
| Recombinant DNA reagent | pGEX-2TK-Nup116$_{(348-458)}$F > A | *Onischenko et al., 2017* | pKW2908, ID351 | *E. coli* expression of GST-tagged mutant GLFG domain of yeast Nup116 (human Nup98 homologue) |
| Recombinant DNA reagent | pGEX-2TK Nsp1$_{(497-609)}$ | *Yamada et al., 2010* | pKW1609; pK1100 | *E. coli* expression of truncated, GST-tagged FG domain of yeast Nsp1 (human Nup62 homologue) |
| Antibody | Anti-TDP-43 (3H8) (mouse monoclonal) | Abcam | Cat#: ab104223 | WB (1:2000) |
| Antibody | Anti-Matrin 3 (rabbit monoclonal) | Abcam | Cat#: ab151714 | WB (1:5000) |
| Antibody | Anti-importin α (mouse monoclonal) | BD Bioscience | Cat#: 610485 | WB (1:2000) |
| Antibody | Anti-transportin 1 (mouse monoclonal) | BD Bioscience | Cat#: 558660 | WB (1:1000) |
| Antibody | Anti-Ran (mouse monoclonal) | BD Bioscience | BD:610341 | WB (1:500) |
| Antibody | Anti-FUS (rabbit polyclonal) | Bethyl | Cat#: A300-302A | WB (1:1000) |
| Antibody | Goat anti-rat (goat polyclonal, HRP linked) | Cell Signaling Technology | Cat#: 7077S | WB (1:5000) |
| Antibody | Goat anti-rabbit (goat polyclonal, HRP-linked) | Cell Signaling Technology | Cat#: 7074S | WB (1:5000) |
| Antibody | Horse anti-mouse (horse polyclonal, HRP-linked) | Cell Signaling Technology | Cat#: 7076S | WB (1:5000) |
| Antibody | Anti-RCC1 (rabbit polyclonal) | GeneTex | Cat#: GTX104590 | WB (1:2000) |
| Antibody | Anti-Nup62 (rat monoclonal) | Millipore Sigma | Cat#: MABE1043 | WB (1:500) |
| Antibody | Anti-DDX3X (rabbit polyclonal) | Millipore Sigma | Cat#: HPA001648 | WB (1:1000) |
| Antibody | Anti-importin β (mouse monoclonal) | Millipore Sigma | Cat#: I2534 | WB (1:2000) |
| Antibody | Anti-Nup54 (rabbit polyclonal) | Millipore Sigma | Cat#: HPA035929 | WB (1:250) |
| Antibody | Anti-ribosomal protein 6 (RPS6) (mouse monoclonal) | Santa Cruz Biotechnology | Cat#: sc-74459 | WB (1:1000) |
| Antibody | Anti-Nup133 (mouse monoclonal) | Santa Cruz Biotechnology | Cat#: sc-376699 | WB (1:2500) |
| Antibody | Anti-RanGAP1 (C-5) (mouse monoclonal) | Santa Cruz Biotechnology | Cat#: sc-28322 | WB (1:50) |

*Continued on next page*

*Continued*

| Reagent type (species) or resource | Designation | Source or reference | Identifiers | Additional information |
|---|---|---|---|---|
| Antibody | Anti-Nup98 (2H10) (rat monoclonal) | Santa Cruz Biotechnology | Cat#: sc-101546 | WB (1:2000) |
| Antibody | Anti-hnRNP A1 (4B10) (mouse monoclonal) | Santa Cruz Biotechnology | Cat#: sc-32301 | WB (1:200) |
| Antibody | Anti-hnRNP A2/B1 (EF-67) (mouse monoclonal) | Santa Cruz Biotechnology | Cat#: sc-53531 | WB (1:200) |
| Commercial assay or kit | miRNeasy kit | Qiagen | Cat#:217004 | |
| Chemical compound, drug | Sypro Ruby Protein Gel Stain | Millipore Sigma | Cat#: S4942 | |
| Chemical compound, drug | SYBR Gold Nucleic Acid Stain | ThermoFisher Scientific | Cat#: S11494 | |
| Chemical compound, drug | Importazole | Millipore Sigma | Cat#: SML0341 | |
| Chemical compound, drug | Alexa Fluor 647 NHS ester | ThermoFisher Scientific | Cat#: A37573 | |
| Chemical compound, drug | Alexa Fluor 488 C5 maleimide | ThermoFisher Scientific | Cat#: A10254 | |
| Chemical compound, drug | EZ-Link Sulfo-NHS-LC-biotin No-weigh format | ThermoFisher Scientific | Cat#: A39257 | |
| Chemical compound, drug | Ni-NTA Agarose | Qiagen | Cat#: 30210 | |
| Chemical compound, drug | Glutathione Sepharose 4B | GE Healthcare | Cat#:17-0756-01 | |
| Chemical compound, drug | Dextran, Texas Red, 10,000 MW | ThermoFisher Scientific | Cat#: D1863 | |
| Chemical compound, drug | Dextran, Texas Red, 40,000 MW | ThermoFisher Scientific | Cat#: D1829 | |
| Chemical compound, drug | Dextran, Texas Red, 70,000 MW | ThermoFisher Scientific | Cat#: D1864 | |
| Chemical compound, drug | Digitonin, high purity | Calbiochem | Cat#: 300410 | |
| Other | Ribo Ruler High Range RNA ladder | ThermoFisher Scientific | Cat#: SM1821 | |
| Other | HIS-Select HF Nickel Affinity Gel | Millipore Sigma | Cat#: HD537 | |
| Other | DNase (RNase free) | Qiagen | Cat#: 79254 | |
| Other | RNase A | ThermoFisher Scientific | Cat#: EN0531 | |

*Continued on next page*

*Continued*

| Reagent type (species) or resource | Designation | Source or reference | Identifiers | Additional information |
| --- | --- | --- | --- | --- |
| Other | Glutathione-coated polystyrene particles 6.0–8.0 | Spherotech | Cat#: GSHP-60–5 | |
| Other | Neutravidin-coated polystyrene particles 6.0–8.0 | Spherotech | Cat#: NVP-60–5 | |
| Other | GFP Trap Magnetic Agarose | Chromotek | Cat#: Gtma-20 | |
| Other | Wheat germ agglutinin | Millipore Sigma | Cat#: L0636 | |
| Other | BSA, fatty acid-free | Roche | Cat#: 03117 057001 | |

## DPR synthesis

10-and 20-mer dipeptide repeat proteins with C-terminal lysine (for solubility) and cysteine (for fluorescent tagging, that is GPGPGPGPGPGPGPGPGPGPKC) were synthesized by Genscript (Nanjing, China) and 21st Century Biochemicals (Marlborough, MA) and verified by mass spectrometry to be free of trifluoroacetic acid adducts. Lyophilized powder was diluted in 0.1x XB' buffer (5 mM sucrose, 10 mM KCl, 1 mM HEPES, pH 7.7) and frozen in single use 10 mM aliquots at −80°C after snap freezing in liquid nitrogen.

## Cloning of recombinant constructs

Restriction cloning was used to insert the ORF from pQE-ZZ-RanQ69L (*Nachury and Weis, 1999*) between the BamH1 and HindIII sites in pRSET A, resulting in pRSET ZZ-RanQ69L. The pRSET zzRCC1 was created by inserting the PCR-amplified wild-type (WT) human RCC1 C-terminally of the ZZ-tag in pRSET A. Site-directed mutagenesis and PCR cloning were used to modify Rango-2 (*Kaláb and Soderholm, 2010*) by removing the KPN1 sites from YPet and CyPet (*Nguyen and Daugherty, 2005*) and replacing the Snurportin-1 IBB with the IBB amplified from human importin α1 (KPNA2). While doing so, the IBB-importin α1 domain was inserted either with (pK44) or without (pK188) flexible GGCGG linkers added between the 5' and 3' ends of IBB and the fluorophores. Restriction cloning was used to combine the C-terminal biotin acceptor peptide tag Avitag (GLNDIFEAQKIEWHE) from pAC-6 (Avidity, Aurora, CO) with WT human importin β (*Chi et al., 1997*) in pRSET A vector, resulting in in pRSET importin β-Avitag (pKW1982; pK1099). Restriction cloning in the modified pRSET A with C-terminal Avitag was used to create pRSET-EGFP-Avitag (pK803). The pGEX-2TK1 plasmid for the expression of the *S. cerevisiae* Nsp1(497-608) FxFG domain was obtained from M. Rexach (*Yamada et al., 2010*).

## Recombinant protein expression

Unless otherwise specified, recombinant proteins were expressed in *E. coli* BL21(DE3) cells (Thermo-Fisher, Waltham, MA) that were cultured in 1L batches of LB media contained in 2.8L baffle-free Fehrnbach flasks. Protein expression was induced with 0.3 mM IPTG. Centrifugation was used to collect the cells and wash them in the ice-cold protein-specific buffer, as indicated below. Unless otherwise specified, all buffers were pH 7.4. The washed cell pellets were flash-frozen in liquid nitrogen and stored at −80°C, and lysed in ice-cold conditions and in the presence of protease inhibitors, using French pressure cell or microfluidizers. After dialysis in the protein-specific buffer, protein concentration was measured with the Bradford assay (BioRad, Hercules, CA), and single-use aliquots of all proteins were stored at −80°C after flash-freezing in liquid nitrogen.

### Recombinant proteins with GFP-derived tags

For expression of proteins containing GFP variants, including Rango (pK44 and pK188), YFP-M9-CFP, GST-GFP-NLS, and GFP-Avitag, the cells were first outgrown at 37°C until reaching

$OD_{600nm}$ = 0.1–0.3. The cultures were cooled to room temperature (22–25°C), and protein expression was induced at 22–25°C for 12–14 hr.

Cells expressing 6His-tagged fluorescent proteins (Rango pK44 and pK188, YFP-M9-CFP, and GFP-Avitag) were washed and lysed in 10 mM imidazole/PBS and purified with either Ni-NTA agarose (Qiagen, Venlo, Netherlands) or HIS-Select HF Nickel Affinity Gel (Millipore Sigma, St. Louis, MO). The lysates were clarified (40 min, 16000 g, 4°C) and incubated with Ni resin (30–60 min, 4°C). The resin was placed into small chromatography columns, washed with ice-cold 10 mM imidazole/PBS, and the proteins eluted with increasing concentration of imidazole/PBS (25–300 mM). SDS-PAGE was used to select and pool batches with the highest purity, prior to dialysis in PBS or XB buffer (50 mM sucrose, 100 mM KCl, 10 mM HEPES, 0.1 mM $CaCl_2$, 1 mM $MgCl_2$, pH 7.7).

Cells expressing GST-GFP-NLS were washed and lysed with TBSE (50 mM Tris, 150 mM NaCl, 4 mM EDTA, pH8.0), the lysate clarified, and the protein affinity-purified on glutathione sepharose (Roche, Basel, Switzerland). After washes with TBSE, the proteins were eluted with TBSE containing increasing concentrations of glutathione (2.5–10 mM). Proteins eluted with 2.5 and 5 mM glutathione were pooled and dialyzed in PBS before storage.

## FRET assay mix with importin β and rango

Full-length human importin β was expressed from pET30a-WT importin β (pKW485; *Chi et al., 1997*) at the Protein Expression Laboratory (PEL, National Cancer Institute, Frederick, MD). The transformed BL21DE3 cells were grown at 37°C in an 80L Bioflow 500 bioreactor (New Brunswick Scientific, Edison, NJ) until $OD600_{nm}$ = 0.6, cooled to 22°C and the expression was induced with 0.3 mM IPTG. After 12 hr induction, cells were harvested with the CARR continuous flow centrifuge and lysed in PBS with 10 mM imidazole and 5 mM TCEP with a 110EH Microfluidizer (Microfluidics, Westwood, MA) using 2 passes at 10,000 PSI under chilled conditions. The lysates were flash-frozen in liquid nitrogen and stored at −80°C. After thawing, the lysates were clarified and incubated with HIS-Select HF Nickel Affinity beads. The beads were washed with 10 mM imidazole/PBS and the protein eluted with 200 mM imidazole/PBS before dialysis in XB. The purified importin β was combined with a freshly thawed aliquot of Rango2-α1 (pK188) at 2.5:1 molar ratio ratio (12.5 μM importin β, 5 μM Rango) and supplemented with 3% glycerol. Measurement of Rango fluorescence emission in a spectrometer (see below) was used to verify the FRET-off state of the Rango/importin β mixture before freezing.

## FRET assay mix with zz-RCC1 and zz-RanQ69L

The expression of zz-RCC1 was induced at $OD_{600nm}$ = 0.4, followed by incubation at 22°C for 4 hr. The cells were washed with PBS, 10 mM Imidazole, 1 mM $MgCl_2$, 5 mM TCEP, 0.2 mM AEBSF, pH 8.0, and lysed by ice-cold microfluidizer. The clarified lysates were used to isolate the zz-RCC1 proteins on HIS-Select HF Nickel Affinity beads, as described above. Proteins eluted with 0.2M imidazole/PBS were dialyzed in PBS before storage. The zz-RanQ69L (pKW1234; pK1097) was expressed from BL21DE3 cells at PEL in an 80L bioreactor, using conditions described for importin β above, except that expression was induced with 0.3 mM IPTG at 37°C for 3 hr, and lysis was performed in PBS with 10 mM Imidazole, 5 mM TCEP, 2 mM $MgCl_2$, and 1 mM GTP. The lysates were clarified and bound to HIS-Select HF Nickel Affinity beads (Millipore Sigma). The Ni resin was washed with ice-cold 10 mM imidazole/PBS and the protein eluted with 0.2M imidazole/PBS, followed by dialysis in XB. After measuring the concentration, 60 μM zzRCC1 and 2.4 μM zzRCC1 were combined in XB containing 2 mM GTP. Before aliquoting and storage, the measurement of Rango fluorescence emission in a spectrometer (see below) was used to verify that the zzRanQ69L-GTP-containing mix robustly induced Rango dissociation from importin β.

## Importin β biotinylation

To prepare biotinylated WT importin β-Avitag (pKW1982, pK1099), BL21DE3 cells (New England Biolabs, Ipswich, MA) were co-transfected with the respective plasmids together with pAC-biotin ligase (Avidity), followed by plating and growth in LB media containing ampicillin and chloramphenicol. After the 37°C cultures reached $OD_{600nm}$ + 0.4–0.6, the cultures were cooled to room temperature, supplemented with 100 μM D-biotin, and the expression was induced with 0.3 mM IPTG at

room temperature for 8–11 hr (pKW762). Proteins were purified on Ni-NTA resin as described for the non-biotinylated importin β fragments.

## FG- and GLFG-nucleoporin fragments

The expression of GST- Nsp1$_{(497-609)}$ in BL21(DE3) cells grown in LB media was induced at OD$_{600nm}$ = 0.4–0.6, followed by incubation at 37°C for 3–5 hr. The GST-tagged *S. cerevisiae* pGEX-Nup100$_{(1-307)}$, Nup100$_{(1-610)}$, Nup116$_{(348-458)}$ and Nup116$_{(348-458)}$F > A were expressed in T7 Shuffle cells (NEB) that were grown in Dynamite media (*Taylor et al., 2017*) until OD$_{600nm}$ = 0.9 before induction with IPTG at 37°C for 3 hr. All the GST-tagged Nup fragments were purified using glutathione-sepharose affinity chromatography, as described for the GST-GFP-NLS above, and dialyzed into PBS before storage.

## Importin β labeling with Alexa-647 and BSA biotinylation

Purified WT importin β (pKW485) diluted to 10 µM in XB was combined with 10-molar excess of Alexa Fluor 647 NHS ester (ThermoFisher), using freshly-prepared 10 mM dye in anhydrous DMSO. After incubation on ice for 2 hr, the sample was dialyzed in PBS and concentrated on 30kD MWCO filter (Millipore Sigma) before storage. A similar protocol was used to label BSA with 10-molar excess Sulfo-NHS-LC-biotin (ThermoFisher), followed by dialysis in PBS.

## DPR labeling with Alexa-488-maleimide

Just before labeling, Alexa Fluor 488 C5-maleimide (ThermoFisher) was diluted to 20 mM in anhydrous DMSO and further diluted to 1.6 mM in XB' buffer. Freshly thawed 10 mM aliquots of DPRs were diluted to 2 mM with 0.1x XB' and combined with an equal volume of 1.6 mM Alexa Fluor 488 C5-maleimide and kept overnight at 4°C. The unreacted maleimide was quenched by 1:50 (v/v) 100 mM DTT before aliquoting and storage at −80°C. To verify labeling, DPRs were separated by SDS-PAGE followed by fluorescence detection.

## Rango FRET detection

FRET assays for DPR-induced dissociation of the importin β-Rango complex were performed using a mix of 5 µM Rango-2/α1 (pK188) and 12.5 µM importin β (pKW485) prepared as described above. After thawing on ice, the mix was diluted to 20 nM Rango and 50 nM importin β in TBS, pH 7.4, 0.01% Tween-20 (TTBS), supplemented with increasing DPR concentrations, and mixed by brief vortexing at low speed. The positive control reactions for RanGTP-induced Rango-importin β dissociation were prepared by adding increasing concentrations of ZZ-RanQ69L-GTP to the samples, using a freshly thawed aliquot of 60 µM ZZ-RanQ69L, 2.4 µM ZZ-RCC1, 2 mM GTP in XB. The assay buffer alone was used as a blank. A Fluoromax-2 spectrometer (Jobin Yvon Horiba, Piscataway, NJ) was used to detect the Rango emission spectra (460–550 nm, in 1 nm increments) while exciting the samples at 435 nm. The excitation and emission bandpass were set to 5 nm and integration time to 0.05 s. Peak emissions were recorded at 480 nm (donor) and 535 nm (acceptor) in all samples, and background emission subtracted at the same wavelengths in the blank. The FRET signal was calculated as the ratio of background-subtracted acceptor/donor emissions. The signal detected in the untreated sample (20 nM Rango and 50 nM importin β-only, the lowest FRET), was then subtracted from the resulting values. Prism v6 (Graphpad, San Diego, CA) was used to calculate the non-linear fit with one site-specific binding model while using the D'Agostino and Pearson K2 test to verify the normality of residuals and the Runs test to assure non-significant deviation from the model.

## Biochemical pulldown assay for DPR-induced Rango-importin β dissociation

An aliquot of 5 µM Rango-2/α1 + 12.5 µM importin β mix was diluted to 20 nM Rango and 50 nM importin β in TTBS, supplemented with increasing concentrations of GR10 or PR10, mixed by vortexing, and incubated for 30 min at room temperature. GFP-Trap magnetic beads (Chromotek, Planegg-Martinsried, Germany) were washed and resuspended in TTBS. At the end of incubation, 8 µl bead suspension was added to each sample and mixed by rotation for 15 min. The supernatant was removed and beads washed 3 times with TTBS before boiling in 20 µl SDS-PAGE sample buffer with 2% β-mercaptoethanol. Samples were separated by SDS-PAGE and anti-GFP western blot

performed as detailed below to detect Rango. After detecting the ECL signal, membranes were stained with Coomassie Brilliant R250 to detect importin β. Background-subtracted signals were determined by Image Lab 6.01 (BioRad) and the Rango ECL signal normalized to the importin β signal within each lane.

## Electrophoretic mobility gel shift assay for RNA-DPR interaction

Aliquots of total HEK RNA (3 µg) were mixed with either 4 µl 50 µM DPR-AF488 in 0.1x XB' or with 1 µl 0.2% SYBR Gold nucleic acid stain (ThermoFisher) diluted in water. After 5 min incubation at room temperature, the samples were supplemented with Fast Digest loading buffer (ThermoFisher; no nucleic acid stain) and separated by electrophoresis on native 1% agarose gel in TBE, alongside with lanes containing HEK RNA (3 µg) or RNA ladder mixed with SYBR Gold. Immediately after electrophoresis, the gels were photographed with Bio-Rad ChemiDoc XRS+ using UV transillumination to simultaneously visualize the AF488-labeled R-DPRs and SYBR Gold-labeled RNA signals (where added).

## Bead halo assay

The bead halo assay was carried out as described with minor modifications (*Patel and Rexach, 2008*), using 6–8 µM polystyrene beads coated with neutravidin (for biotinylated proteins) or gluta-thione (for GST-fusion proteins) (Spherotech, Lake Forest, IL). Beads were coated overnight at 4°C at saturating concentrations per manufacturers' instructions and rinsed 2x in binding buffer (20 mM HEPES [pH 7.4], 150 mM KOAc, 2 mM Mg(OAc)$_2$, 1 mM DTT, 0.1% Tween-20). Immediately prior to the assay, fluorescent bait proteins and beads were combined with 4x assay buffer (40 mM EDTA, 40 mg/ml BSA, 500 mM NaCl, and 0.2% Tween) to a total of 40 µL per well, in optical glass-bottom 96-well plates (Cellvis, Mountain View, CA). Reactions were allowed to equilibrate at room temperature for a minimum of 30 min prior to imaging at 100x on an LSM800 confocal microscope (Zeiss, Oberkochen, Germany). Intensity profiles comparing the maximum rim intensity to the background were plotted in ImageJ (NIH) by an investigator blinded to experimental conditions.

## Mouse primary cortical neuron culture and permeabilization

All animal procedures were approved by the Johns Hopkins Animal Care and Use Committee. Timed pregnant C57BL/6J females (Jackson Laboratory, Bar Harbor, ME) were sacrificed by cervical dislocation at E16, cortex dissociated, and cells plated at 50,000/well on poly-D-lysine/laminin-coated, optical glass-bottom 96-well plates. Growth medium consisted of Neurobasal supplemented with B27, Glutamax, and penicillin/streptomycin (Gibco/ThermoFisher). At 5–7 days in vitro, neurons were rinsed in prewarmed PBS and permeabilized for 4 min. at 37° in a hypotonic solution containing 0–40 µM Tris-HCl pH 7.5 (to cause osmotic swelling) and 50–150 mg/ml BSA (for molecular crowding/mechanical support). Following permeabilization, cells were placed on ice and rinsed 2 × 5 min in transport buffer (TRB, 20 mM HEPES, 110 mM KOAc, 2 mM Mg(OAc)$_2$, 5 mM NaOAc, 0.5 mM EGTA, 250 mM sucrose, pH 7.3, with protease inhibitor cocktail). All rinse and assay buffers were supplemented with 50 mg/mL BSA. The optimal hypotonic buffer and BSA concentration varied by batch, and was optimized prior to each set of assays for ability to permeabilize the majority of plasma membranes while maintaining nuclear exclusion of a 70 kD fluorescent dextran (ThermoFisher).

## HeLa cell culture and permeabilization

A single cell-derived clone of HeLa cells (ATCC, Manassas, VA; mycoplasma negative and validated by STR profiling) were maintained in OptiMEM (Gibco/ThermoFisher) with 4% FBS and plated on uncoated optical glass-bottom 96 well plates, at appropriate densities to reach 70–90% confluence on the day of the transport assay. To permeabilize, cells were rinsed for 2 min in ice-cold PBS, and permeabilized on ice for 10 min in 15–30 µg/mL digitonin (Calbiochem, San Diego, CA) in permeabilization buffer (PRB, 20 mM HEPES, 110 mM KOAc, 5 mM Mg(OAc)2, 0.5 mM EGTA, 250 mM sucrose, pH 7.5, with protease inhibitor cocktail). Following permeabilization, cells were placed on ice and rinsed 3 × 5 min in transport buffer (TRB, 20 mM HEPES, 110 mM KOAc, 2 mM Mg(OAc)$_2$, 5 mM NaOAc, 0.5 mM EGTA, 250 mM sucrose, pH 7.3, with protease inhibitor cocktail). The optimal digitonin concentration varied by cell density and passage number, and was optimized prior to each

set of assays for the ability to permeabilize the majority of plasma membranes while maintaining nuclear exclusion of a 70 kD fluorescent dextran (ThermoFisher).

## Nuclear import assays

### Assay components

Nuclear import was carried out essentially as described (Adam et al., 1990) with modified sucrose-containing buffers (Zhu et al., 2016). Concentrated whole cell lysates were prepared from HEK293T cells (ATCC, mycoplasma negative and validated by STR profiling), grown in 150 mm dishes and sonicated on ice in 1X TRB in the presence of protease inhibitor cocktail (Roche). The lysates were clarified (15 min, 14000 g, 4C), snap frozen in liquid nitrogen, and stored in single use aliquots at −80C. Total HEK cell RNA was extracted using miRNEasy kits according to the manufacturers' protocol, with DNase digestion (Qiagen). RNA concentration was measured by Nanodrop (ThermoFisher), and all 260/280 ratios were verified to be > 2.0. Energy regeneration (ER) mix consisted of 100 µM ATP, 100 µM GTP, 4 mM creatine phosphate, and 20 U/mL creatine kinase (Roche).

### Standard assay setup

Reaction mixes consisting of 2.5 mg/ml lysate, ER, fluorescent cargo (200 nM Rango and YFP-M9-CFP, 500 nM GST-GFP-NLS), Hoechst, DPRs, RNA, or inhibitors (100 µM importazole (IPZ, Millipore Sigma); 0.8 mg/mL wheat germ agglutinin (WGA, Millipore Sigma) were assembled on ice during cell permeabilization. DPRs or inhibitors were allowed to equilibrate in cell lysate for at least 30 min prior to initiation of transport. Every plate included: (1) Cargo alone: fluorescent cargo, but no ER or lysate, (2) Untreated controls: fluorescent cargo, ER, and lysate, and (3) Inhibitor: fluorescent cargo, ER, lysate, and IPZ (Rango and GST-GFP-NLS reactions) or WGA (YFP-M9-CFP). Preassembled transport reactions were then transferred onto permeabilized cells via multichannel pipette, and allowed to proceed at room temperature for 2 hr (Rango, YFP-M9-CFP) or 4 hr (GST-GFP-NLS). Cells were fixed in 4% paraformaldehyde/PBS, rinsed 2x with PBS, and transferred to 50% glycerol/PBS for immediate imaging.

### Variations

For a subset of neuron transport assays, transport was monitored live via time lapse imaging every 5 min for 30 min. In a subset of HeLa assays, transport reactions were centrifuged before use at 14,000g × 15 min to separate soluble and insoluble fractions. In another variation, the transport lysate + DPR and ER mix was allowed to preincubate on the permeabilized HeLa cells for 1 hour prior to initiation of transport, rinsed 1x with TRB, and transport initiated with fresh lysate, cargo, and ER.

### Imaging and data analysis

Multiple non-overlapping fields per well (4 for time-lapse imaging, 9-16 for fixed imaging), were captured at 40x on an ImageXpress Micro XLS high-content microscope (Molecular Devices, San Jose, CA), and the ratio of nuclear to cytoplasmic fluorescence intensity was calculated using the MetaXpress automated translocation-enhanced module. Raw data were filtered to exclude autofluorescence and the mean N/C ratio from wells without ER or cell lysate was subtracted from all values. Resulting N/C ratios were expressed as % untreated, to permit comparisons across biological replicates.

## Passive import assays

HeLa cells were permeabilized as above, rinsed 3x with TRB, and reaction mix containing 2.5 mg/ml HEK lysate, ER (to mimic the active import conditions), Hoechst, and/or DPRs were added and allowed to preincubate directly on the permeabilized cells for at least 30 min. 0.8 mg/ml WGA was used as a positive control. Cells were mounted on a Zeiss LSM800 confocal microscope, reaction mix withdrawn, and immediately replaced with fluorescent dextran (ThermoFisher) or recombinant GFP (pK803) in fresh lysate/ER mix to initiate the passive import reaction. A single 40x frame (containing 20–30 cells/well) was imaged per well, with images collected every 10 s for 5 min. The ratio of nuclear fluorescence intensity to local background at each timepoint was analyzed using Imaris

(Bitplane, Zurich, Switzerland), and values for each cell were expressed as a ratio of time 0 (1 = no influx, >1 = influx).

## Mass spectrometry

50 µM GR10 or 25 µM PR10 (in duplicate) were added to 5 mg/ml HEK whole cell lysate (in TRB with ER), incubated for 60 min at 37°C and aggregates were pelleted by centrifugation at 16,000 g for 10 min. Supernatants were removed and pellets washed 2x and resuspended in $MgCl_2$- and $CaCl_2$- free DPBS (ThermoFisher), then flash-frozen in liquid nitrogen and stored at −80°C before further processing and analysis by the Johns Hopkins Mass Spectrometry and Proteomics core facility. Pellets were reduced/alkylated with DTT/IAA, reconstituted in TEAB/acetonitrile, and sonicated for 15 min prior to overnight digestion with Trypsin/LysC (Promega, Madison, WI) at 37°C. Some precipitate remained; supernatants were desalted and analyzed by LC/MS/MS on a QExactive_Plus mass spectrometer (ThermoFisher). MS/MS spectra were searched via Proteome Discoverer 2.2 (ThermoFisher) with Mascot 2.6.2 (Matrix Science, London, UK) against the RefSeq2017_83_ human species database (NCBI). Protein probabilities were assigned by the Protein Prophet algorithm (*Nesvizhskii et al., 2003*). Protein identifications were accepted if they contained at least 2 identified peptides at false discovery rate less than 1.0%. Gene ontology analysis was carried out using the DAVID algorithm v6.8 (May 2016, https://david.ncifcrf.gov/; *Huang et al., 2009a*; *Huang et al., 2009b*). The mass spectrometry data have been uploaded to the ProteomeXchange Consortium (http://proteomecentral.proteomexchange.org) via the PRIDE partner repository (*Vizcaíno et al., 2013*) with the dataset identifier PXD015656 and 10.6019/PXD015656.

## DPR aggregation assay and western blots

Supernatant and pellet fractions (for all 5 DPRs, and control/buffer only) were prepared as in the nuclear transport assays, by adding 100 µM 10mers to 2.5 mg/ml HEK lysate in 100 µL TRB. Supernatants were boiled in Laemmli (BioRad) for 5 min. Pellets were boiled for 15 min followed by sonication in order to fully disperse aggregates for SDS-PAGE. Equal volumes of supernatant and pellet fractions were run on 4–12% Bolt Bis-Tris Plus gels (ThermoFisher), transferred to nitrocellulose membrane using an iBlot2 dry blotting system (ThermoFisher). Protein loading was analyzed by BLOT-Faststain (G-biosciences, St. Louis, MO), according to the manufacterers' instructions. For immunodetection, membranes were blocked with 5% non-fat milk in TBST and probed by sequential incubation with the primary antibodies as detailed in the table below. Detection was by HRP-conjugated secondary antibodies/chemiluminescence using an ImageQuant LAS 4000 system (GE, Chicago, IL). To permit sequential probing of membranes without stripping, signals were quenched by incubation with prewarmed 30% $H_2O_2$ for 20 min (*Sennepin et al., 2009*). Band intensities were measured by ImageQuant software. For pellet vs. supernatant fractions, all were expressed as percent untreated control. For blots in *Figure 1E*, samples were run on 4–20% SDS PAGE minigels (ThermoFisher) and blotted to PVDF membranes (Immun-Blot PVDF, Bio-Rad) using the Bio-Rad TransBlot Turbo apparatus, and probed as above. The chemiluminescence signal was captured with a Bio-Rad ChemiDoc XRS+ digital imaging system.

## Statistical analysis

Data analysis, graphing, and statistical analyses were carried out using Prism v6-v8 (Graphpad), according to methods detailed under each experimental approach above and in the figure legends.

## Acknowledgements

We thank all members of the Rothstein lab for helpful discussion. Lin Xue and Svetlana Vidensky provided expert technical assistance. Robert Cole and Tatiana Boronina in the Johns Hopkins Proteomics Core assisted with proteomics analysis and interpretation. We gratefully acknowledge M Rexach, M Dasso, E Onischenko and K Weis for providing expression plasmids.

## Additional information

### Funding

| Funder | Grant reference number | Author |
|---|---|---|
| National Institute of Neurological Disorders and Stroke | K08NS104273 | Lindsey R. Hayes |
| National Institute of Neurological Disorders and Stroke | R01NS094239 | Jeffrey D Rothstein |
| National Institute of Neurological Disorders and Stroke | P01NS099114 | Jeffrey D Rothstein |
| National Institute on Aging | RF1AG062171 | Jeffrey D Rothstein |

The funders had no role in study design, data collection and interpretation, or the decision to submit the work for publication.

### Author contributions

Lindsey R Hayes, Conceptualization, Methodology, Validation, Formal analysis, Investigation, Data curation, Writing-original draft preparation, Writing-review & editing, Visualization, Funding acquisition; Lauren Duan, Investigation, Formal analysis, Validation, Writing-review & editing; Kelly Bowen, Investigation, Formal analysis, Writing-review & editing; Petr Kalab, Conceptualization, Methodology, Validation, Formal analysis, Investigation, Resources, Data curation, Writing-review & editing, Visualization, Supervision; Jeffrey D Rothstein, Conceptualization, Writing-review & editing, Supervision, Project administration, Funding acquisition

### Author ORCIDs

Lindsey R Hayes (iD) https://orcid.org/0000-0003-4818-6054
Petr Kalab (iD) https://orcid.org/0000-0002-8145-7728

### Ethics

Animal experimentation: All animal studies were approved by the Johns Hopkins University Animal Care and Use Committee (protocol MO17M75), and carried out in strict accordance with the Guide for the Care and Use of Laboratory Animals of the National Institutes of Health.

### Decision letter and Author response

Decision letter https://doi.org/10.7554/eLife.51685.sa1
Author response https://doi.org/10.7554/eLife.51685.sa2

## Additional files

### Supplementary files

• Transparent reporting form

### Data availability

The mass spectrometry data have been uploaded to the ProteomeXchange Consortium (http://proteomecentral.proteomexchange.org) via the PRIDE partner repository (Vizcaino et al., 2013) with the dataset identifier PXD015656 and https://doi.org/10.6019/PXD015656.

The following dataset was generated:

| Author(s) | Year | Dataset title | Dataset URL | Database and Identifier |
|---|---|---|---|---|
| Hayes, Duan, Bowen, Kalab, Rothstein | 2020 | C9orf72 arginine-rich dipeptide repeat proteins disrupt karyopherin-mediated nuclear import | https://doi.org/10.6019/PXD015656 | ProteomeXchange Consortium, PXD015656 |

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
