## [Decision Letter]

**Acceptance summary:**

We recognize that you and your colleagues have taken the earlier advice from our reviewers seriously. We believe that the revised manuscript represents a sophisticated set of approaches to identify how the two arginine containing polydipetide repeat proteins encoded by the ALS-causing repeat expansion in *C9orf72* affect nuclear import and will be of interest to a broad audience.

**Decision letter after peer review:**

Thank you for submitting your article "*C9orf72* arginine-rich dipeptide repeat proteins disrupt importin β-mediated nuclear import" for consideration by *eLife*. Your article has been reviewed by three peer reviewers, one of whom is a member of our Board of Reviewing Editors, and the evaluation has been overseen Huda Zoghbi as the Senior Editor. The reviewers have opted to remain anonymous.

The reviewers have discussed the reviews with one another and the Reviewing Editor has drafted this decision to help you prepare a revised submission.

Summary:

The reviewers agree that your work represents a comprehensive, sophisticated, well-designed set of experimental efforts that test how the arginine containing polydipeptide repeat (R-DPRs) proteins produced in an inherited form of ALS affect nucleocytoplasmic transport. All reviewers agreed that the FRET and biochemical assays, including in situ permeabilized cell reconstitutions, do establish a direct interaction with importin β which disrupts its cargo loading, and correspondingly, inhibiting nuclear import in a manner that can be diminished by increasing RNA concentration (with the RNA presumably competing with importin β for R-DPR binding). Additionally, your efforts support that transport inhibition is not through sequestration of nuclear import components or blockage of the nuclear pore complex. One reviewer reported that the text and figures have been beautifully prepared, producing a compelling manuscript that was a pleasure to read.

That said, there are areas where we would like further clarification.

Essential revisions:

1) Your effort begins by probing a link between R-DPRs and importin β, and the first few figures appear to confirm this link as being important for interruption of NCT in *C9ALS*. However, subsequent experiments argue against this mechanism, and indicate that the action of GR differs from that of PR. By the end, we felt as if the original hypothesis had been disproved to some degree, but an alternative hypothesis had not been presented or adequately explored. The result was some confusion over what, precisely, R-DPRs are doing and how their actions result in NCT impairment.

2) The experiments raise several questions that are not addressed in the text or in subsequent investigations. These include:

a) The modest effect of PA and GA upon NCT (Figure 1K);

b) The inhibition of Trp1-mediated NCT (Figure 1L) which argues against importin β as a mechanism of action; the inconsistent effects of R-DPR length, particularly with regards to the Rango reporter (Figure 1—figure supplement 2B, C);

c) The troublesome nonspecific binding of GR/PR in Figures 2 and 3;

d) The increase in PR binding with addition of lysate (Figure 2);

e) The apparent increase in passive NCT flux with PR/GR (Figure 3); and

f) The isolation of more proteins by GR than by PR (Figure 4B) despite the more pronounced effect of PR in most assays.

3) The DPRs used in this study are considerably shorter than those observed in patients. Indeed, the purified DPRs used in this study are 10-mers and 20-mers, well within the normal range for non-ALS individuals, and used at μM amounts. Even though there is evidence of toxicity with these repeat sizes in various cellular systems, exploration of key aspects of this mechanism in the context of repeat sizes more reflective of those observed in patients should be considered.

4) The majority of assays are executed in HeLa cells. Given data that there may be important differences in NCT in different cell types, the main claims should be explored in neurons; if not at least discussed.

---

## [Author Response]

Essential revisions:1) Your effort begins by probing a link between R-DPRs and importin β, and the first few figures appear to confirm this link as being important for interruption of NCT in C9ALS. However, subsequent experiments argue against this mechanism, and indicate that the action of GR differs from that of PR. By the end, we felt as if the original hypothesis had been disproved to some degree, but an alternative hypothesis had not been presented or adequately explored. The result was some confusion over what, precisely, R-DPRs are doing and how their actions result in NCT impairment.

Thank you for this feedback. It seems that in trying to be very clear about the model, we oversimplified things, both in terms of the expected differences between GR and PR and the narrow focus on importin β. Indeed, the action of GR clearly differs from PR, throughout our current study and in other recent publications. Moreover, while we focused on the possibility that importin β could be targeted by R-DPRs that mimic its cargoes, we assume that in parallel, R-DPRs could disrupt other nuclear transport receptors (karyopherins), as supported by our transportin (M9) data. All of this would be predicted based on the divergent molecular properties of GR and PR and the role of basic residues in nuclear import complex formation. We have therefore changed the title from “importin β” to “karyopherin” and made major edits throughout the manuscript reflecting both of the above topics.

Based on their structure and published studies, the differing behavior of GR and PR is not surprising. On one hand, both are positively charged (theoretical π = 12.95 for PR10 and GR10), with a predicted disordered state and polynucleotide-binding activity (https://predictprotein.org/; Yachdav et al., 2014). However, because glycine is conformationally flexible and its side chain is comprised of a single hydrogen, while the integrated proline cyclic side-chain induces a rigid bend in the peptide backbone, poly-GR is predicted to accommodate many different macromolecular ligands, while poly-PR may be more selective. As recently reviewed by Freibaum and Taylor, 2017, GR and PR have differing half-lives, differing intracellular localization, and differing properties with regard to stress granule dynamics and mitochondrial function. Moreover, in yeast the modifiers of GR and PR toxicity showed little overlap (Chai and Gitler, 2018).

Accordingly, although both GR and PR showed low nanomolar displacement of importin β from the Rango (IBB) FRET sensor (Figure 1D-FRET), in the presence of cell lysate, PR had greater potency for inhibiting importin- and transportin-mediated nuclear import (Figure 1-permeabilized cell assays), higher relative selectivity for importin β (Figure 2C-D-bead halo assays), and differing behavior with respect to the soluble phase of nuclear import (Figure 5C-permeabilized cell assays). Although not quantitative in this regard, our mass spectrometry results showed 21% of hits uniquely bound GR10, and 11% PR10. This was even higher at 35% (GR50) and 25% (PR50) unique hits in the data set from Lee and colleagues (2016). We have edited the manuscript to discuss the differences between GR and PR in our assays and how that relates to the conclusions. Namely, that although in a purified system both of the R-DPRs show a similar propensity to bind importin β (unlike GP, GA, and PA), in the more complex environment of the cell, our findings predict PR to be the more potent disruptor.

In summary, our results do support our initial hypothesis *vis-à-vis* importin β, although we find that PR acts as a more selective inhibitor relative to GR, and there are broader implications of our findings for other karyopherins.

2) The experiments raise several questions that are not addressed in the text or in subsequent investigations. These include:a) The modest effect of PA and GA upon NCT (Figure 1K);

In permeabilized primary neurons exposed to 100 µM GA10 and PA10, we saw a mild but statistically significant decrease in nuclear enrichment of the Rango import cargo at steady state (2 hours, Figure 1K). This was a late effect (>30 min), as we did not see any separation from controls within the first 30 minutes of nuclear import (data added to Figure 1—figure supplement 1). The potential mechanism and physiologic relevance are not clear, given the high concentrations used and the lack of a similar effect in HeLa cells (Figure 1—figure supplement 2). In both cases the reaction mixture (cell lysate, cargo, and energy) is equivalent, therefore it is possible that high concentrations of GA and PA may somehow react differently with either the neuronal nuclear pore complexes or nuclear contents required for the transport reaction. In more complex in vitroand in vivomodel systems, GA has been proposed to sequester nuclear transport components (Khosravi et al., 2016; Zhang et al., 2016), although we did not observe that in the sedimentation assay when compared to R-DPRs at equivalent concentrations (Figure 4C). We have added additional comments to the Discussion.

b) The inhibition of Trp1-mediated NCT (Figure 1L) which argues against importin-ß as a mechanism of action; the inconsistent effects of R-DPR length, particularly with regards to the Rango reporter (Figure 1—figure supplement 2B, C);

The nuclear import and export pathways, including those that are mediated by transportin and importin β, do not operate in series, but rather simultaneously in parallel, and for a wide repertoire of distinct cargoes. In the revised manuscript, we describe how the known features of importin β/α and transportin NLSs, and the properties of R-DPRs, predict that R-DPRs could employ distinct mechanisms to disrupt cargo loading on both transportin and importin β, in parallel.

In case of transportin cargos, the positive charge of arginines in R-DPRs could promote their electrostatic interactions with the aromatic ring of the side chain of tyrosine (cation-pi interactions) which is the essential part of the PY-NLS of transportin. Since the PY-NLS also contains an N-terminal basic and R-containing region, R-DPRs could at the same bind transportin via cargo mimicry. Both mechanisms may be contributing. Although we focused on mechanistic implications for importin β, we did not want to overlook what we anticipated would be parallel biology affecting the transportin pathway. We have elaborated further on this issue, and have substituted “karyopherin” for “importin β” in the title, to reflect the importance of this observation.

c) The troublesome nonspecific binding of GR/PR in Figures 2 and 3;

The GR and PR peptides indeed showed a high propensity for nonspecific binding, as would be expected for these highly-charged peptides. To mitigate this, the bead halo assays were done at neutral pH with EDTA (to reduce cation-mediated interactions), NaCl (to disrupt weak ionic interactions), and Tween (to disrupt weak hydrophobic interactions). 10 mg/ml (150 µM) BSA was also included as an additional blocking agent. Nevertheless, such measures did not eliminate the nonspecific binding, either in this paradigm or in our attempts to optimize conventional immunoprecipitation assays with these peptides (not shown).

Within the techniques available to us, we found the bead halo assay to be the most reproducible and quantitative approach to account for this nonspecific binding. We have included well-defined positive and negative controls for expected specific binding, and bare beads as well as sham-coated beads (biotinylated BSA in Figure 2, and F→A mutant Nup116 in Figure 3), to quantify the level of nonspecific binding for valid comparison. Based on minor comment 13, we also added Figure 2—figure supplement 2, confirming the ability of free importin β to compete for R-DPR binding in a dose-dependent manner. This further supports the specificity of the binding.

Moreover, the effect of adding cell lysate in Figure 2 (marked by dotted lines in 2A, and graphed in 2C-D), precisely predicts the results of Figure 5 in the permeabilized cell assay. That is, in the context of cell lysate, PR retains its specific binding for importin β, whereas GR is consumed by binding to other targets (even the nonspecific binding goes away).

Additional comments about this have been added to the description of the bead halo results.

d) The increase in PR binding with addition of lysate (Figure 2);

This may be due to recruitment of additional importin β- and PR-binding partners from the cell lysate onto the beads, providing additional sites for AF488-PR to bind. We added this comment to the Results section describing these data.

e) The apparent increase in passive NCT flux with PR/GR (Figure 3); and

The acceleration of passive transport that we observed for PR and GR (but not for GP, GA, or PA, as has been added to Figure 3—figure supplement 1), was indeed surprising. There are several possible scenarios to explain this, outlined below and discussed in the revised manuscript. Please see also further comments under minor comment #14.

Studies in yeast suggest that to pass through the NPC, cargos must overcome collisions with highly mobile, disordered FG domains, particularly the Nup98 homologues Nup100 and Nup116 (Timney et al., 2016). If R-DPRs target these FG domains, as indicated by the findings of Shi and colleagues (2017) and by our bead halo studies (Figure 3—figure supplement 2), this could drastically increase their positive charge or reduce their mobility, thus reducing their gate-keeping function. The passive cargo exclusion properties of the NPC have also been shown to depend on importin β. By superresolution microscopy, Ma and colleagues (2012) showed that when extra importin β accumulates within the NPCs, concentrating inside the peripherally-localized active transport zone, the central passive transport channel widens, thus allowing faster passive cargo flux. Alternatively, when Kapinos et al., 2017, monitored the passive cargo influx rate in permeabilized cells, they showed that when importin•cargo complexes were depleted from the NPC, the pore became functionally leaky. Although our permeabilized cell experiments indicated the R-DPR-induced accumulation of AF647-labeled importin β at the nuclear rim (Figure 5D), concomitant with the transport blockade, these assays did not apply sufficient resolution to conclude if importin β was accumulated in or depleted from NPC channels. Superresolution microscopy studies would be needed to show the precise site of importin β accumulation (cytoplasmic face or within the peripheral or central channel), to test which of the above effects may be contributing. Single molecule tracking of passive transport through NPCs would also aid in definitively establishing the precise effect of R-DPRs on putative opening of the NPC permeability barrier.

Nevertheless, our passive transport data (Figure 3) do not support the notion that R-DPRs induce block to nuclear transport by occluding the NPC transport channels. Such a conclusion is also strongly supported by the results of the “nuclei preincubation” experiment (Figure 5E-F), where the NPCs were exposed to high R-DPR concentration for 1h before the start of the nuclear import assay, and no deceleration of nuclear import was observed.

f) The isolation of more proteins by GR than by PR (Figure 4B) despite the more pronounced effect of PR in most assays.

This is explained by the structural differences in these two R-DPRs as discussed above in #1. Poly-GR is predicted to accommodate many different macromolecular ligands, while poly-PR is expected to be more selective, due to the marked difference in the flexibility of the peptide backbone. Our assays consistently demonstrate that PR has a higher selectivity for importin β in conditions mimicking the complex cellular milieu, consistent with the more pronounced inhibition of nuclear import.

3) The DPRs used in this study are considerably shorter than those observed in patients. Indeed, the purified DPRs used in this study are 10-mers and 20-mers, well within the normal range for non-ALS individuals, and used at μM amounts. Even though there is evidence of toxicity with these repeat sizes in various cellular systems, exploration of key aspects of this mechanism in the context of repeat sizes more reflective of those observed in patients should be considered.

The length of GR and PR in affected individuals remains unknown due to technical challenges in determining protein size. This issue aside, there is no evidence, to our knowledge, that RAN translation of <30 *C9orf72* repeat sequences occurs in neurologically healthy individuals, nor has deposition of GR and PR been reported in individuals with normal *C9orf72* repeat length. Although all individuals possess short C9 RNA repeats, there are no data at this time to support the idea that non-ALS individuals express short DPRs.

The reviewers’ suggestion to try longer repeats is nevertheless very reasonable, and was why we compared 10mers to 20mers in the transport assays in Figure 1. The marked increase in transport inhibition that we observed for 20mers compared to 10mers does suggest that we would see even more potent effects at longer repeat lengths. However for both of the commercial entities we contracted with to make the peptides for this study, even producing 20mers by solid phase synthesis was technically challenging. Our own attempts at purifying recombinant DPRs in *E. coli* was met with significant difficulty. These remain very technically difficult experiments to do, due to the nature of these proteins.

4) The majority of assays are executed in HeLa cells. Given data that there may be important differences in NCT in different cell types, the main claims should be explored in neurons; if not at least discussed.

Certainly, we agree with the critical need to verify the major claims of this paper in neurons. That was the basis for our extensive efforts (shown in Figure 1 and the supplement) to optimize a method for selective permeabilization of neuronal nuclei. When we first encountered difficulty with the fragility of neurons compared to HeLa, we searched the literature and found no convincing reports of this assay in neurons. We then devised a novel method that allowed us to selectively permeabilize neurons without also rupturing the nuclear membrane. To our knowledge, this is the first convincing and reproducible use of the nuclear transport assay in permeabilized neurons.

Upon doing so, we observed that the effects of GR and PR on Rango nuclear import (the direct importin β cargo) were very reminiscent of that seen in HeLa. Having verified this, we did not see a scientific rationale for continuing to duplicate all assays in neurons, which is significantly more time- and resource-intensive.